# Poetry and the Qurʾan: The Use of *tashbīh* Particles in Classical Arabic Texts

Ali Ahmad Hussein 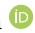

Department of Arabic Language and Literature, University of Haifa, Haifa 3103301, Israel;
ahussein@univ.haifa.ac.il

**Abstract:** This study examines the use of five *tashbīh* (simile) particles which appear in close frequency in pre-and early Islamic poetry and in the Qurʾan. The particles are *ka*-(as), *ka-mā* (such as), *mithl* (like), and derivatives of the roots *ḥsb* (deem) and *shbh* (looks like, similar to). As well as understanding classical Arabic techniques for composition of similes, the study examines aspects of the interrelationship between the Qurʾan and the poetry corpus, the single surviving Arabic text to which the scripture was exposed. It finds greater common structural and lexical similarities between poetry and the Qurʾan in its earlier period (during the Meccan Revelation, 610–622 CE) than later, following the migration of Prophet Muḥammad to Medina (622–632 CE), when other ways of using these particles developed. This suggests surveying these techniques in other texts possibly known to Medinian society, such as the Bible. The present study outlines the premise that qurʾanic composition moved from the influence of the Arabic prototype seen in the poetry in the earliest periods of Revelation to a different form in later periods (texts, possibly biblical). This premise can be further explored by future examination of the interrelationship between the Qurʾan, pre- and early Islamic poetry and the Bible.

**Keywords:** Qurʾan; classical Arabic poetry; rhetoric

## 1. Objectives and Background

In this article, I compare use of *tashbīh* (simile) particles in pre-Islamic poetry and in the Qurʾan. These particles are words that indicate the simile—for example, *ka-mā* (such as). The study uses data gathered by the rhetorical element identifier (REI), a web-based tool developed by Ali Hussein et al.[1] Its database comprises 1908 pre-Islamic poems (22,788 verses, 214,231 words) and the 6200 verses (77,437 words) of qurʾanic text. In its existing form, the REI enables automatic identification of what is known as the "loose or unrestricted simile" (*tashbīh mursal*)—that which contains simile particles,[2] differentiating them from other similes in which the particle does not appear. This type of simile is found frequently, which makes its behaviour in the two corpora significant. In another principal type of simile, with its different sub-types, the particle is not mentioned.[3] Western literature considers this type of simile a metaphor. Its use in pre-Islamic poetry and in the Qurʾan is rarer than the *tashbīh mursal* but its future investigation could prove fruitful.

Let me first clarify four English terms used in this article: simile, tenor, vehicle, and loose/unrestricted. The first connotes the Arabic *tashbīh*, the second and third replace the terms *mushabbah* and *mushabbah bi-hi*, and the fourth is used for *mursal*. I am aware of the problem of employing non-Arabic terms to describe Arabic rhetoric. Hany Rashwan, who has addressed this in detail, argues that different cultures think differently, and each develops its own literary concepts to fit its own language and the orientation of its readers. Arabic rhetorical terms, therefore, describe a type of rhetoric that does not necessarily exist in the same way in other cultures (Hany Rashwan 2020, pp. 335–70). The four Arabic terms used in this article are not, thus, an exact reflection of the non-Arabic terms. I nevertheless bring these translations to give the non-Arabic reader a general idea of what the Arabic term describes. In a sentence like *Zaydun ka-l-asad* (Zayd is like a lion), the *tashbīh* is the

comparison between Zayd and the lion; Zayd is the *mushabbah* and *asad* (a lion) is the *mushabbah bi-hi*.

In its early stages, Arabic rhetoric did not possess a specific term for similes containing particles. It was only later, notably from the eighth/fourteenth century onwards, that these were designated with the term *mursal*.

One of the earliest Arabic rhetoricians, Abū l-Ḥasan al-Rummānī (d. 384/994), distinguishes between the two aforementioned types of simile, one that includes a particle and the other that does not, in the following definition:

> *Wa-lā yakhlū l-tashbīhu min an yakūna fī l-qawli aw fī l-nafs. Fa-ammā l-qawlu fa-naḥwa qawlika: Zaydun shadīdun ka-l-asad. Fa-l-kāf ʿaqadat al-mushabbaha bi-l-mushabbahi bi-hi, wa-ammā l-ʿaqdu fī l-nafsi fa-l-iʿtiqādu li-maʿnā hādhā l-qawl.*

> Every simile can be [conveyed] either in words or in the mind. That which is [conveyed] in words is as when you say: "Zayd is strong like a lion". The [particle] *ka-* (like) connects the tenor (Zayd) to the vehicle (lion). Mental connection, on the other hand, is when you mentally realize this meaning.[4]

According to al-Rummānī, a simile particle in a sentence serves to verbally connect the two parts of the simile (the *mushabbah* and the *mushabbah bi-hi*), providing a clear and direct indication that the sentence contains a simile. In cases where there is no simile particle, as in *Zaydun asadun* (Zayd is a lion), there is no explicit verbal reference to indicate the presence of a simile. However, the comparison is inherently understood by the speaker (or writer) and the listener (or reader), making it evident that Zayd is being likened to a lion.

Similarly, Abū Hilāl al-ʿAskarī (d. after 400/1010) defines the two types of simile as:

> *Al-tashbīh: al-waṣfu bi-anna aḥada l-mawṣūfayni yanūbu manāba l-ākhari bi-adāti l-tashbīhi, nāba manābahu aw lam yanub, wa-qad jāʾa fī l-shiʿri wa-sāʾiri l-kalāmi bi-ghayri adāti l-tashbīh...*

> *tashbīh* uses a simile particle and describes two objects by stating that one substitutes for the other, even when this substitution is not [literally] true. In poetry and various forms of speech, *tashbīh* is occasionally employed without the use of a simile particle...

In this context, the terminology does not make a clear distinction between these two types of similes. The type that employs a particle is considered the primary form and is more commonly used. Abū Hilāl al-ʿAskarī elaborates on this distinction by explaining that one of the two objects in a simile (either the *mushabbah* or the *mushabbah bi-hi*) can both realistically substitute for and fail to substitute for the other. Using the same example as al-Rummānī, he points out that no human possesses the strength of a lion, making it impossible for Zayd to replace an actual lion in reality. In this case, the simile serves to exaggerate the praised quality in the *mushabbah*.[5]

The first rhetorician I encountered who employs the term *tashbīh mursal* is ʿAbd al-Qāhir al-Jurjānī (d. ca. 471/1078), a celebrated grammarian and rhetorician hailing from Gurgān. He delves into the distinctions between simile, metaphor, and a third rhetorical element known as *majāz mursal*, often translated as synecdoche or metonymy. In the first two elements, there exists a relationship of similarity, but this connection is notably absent in the third element:

> *... wa-lā yuʿqalu tashbīhun ḥattā yakūna hāhunā mushabbashun wa-mushabbahun bi-hi, hādhā wa-l-tashbīhu sādhajun mursalun, fa-kayfa idhā kāna ʿalā maʿnā l-mubālaghati, ʿalā an yujʿala l-thānī annahu inqalaba mathalan ilā jinsi l-awwali, fa-ṣāra l-rajulu asadan wa-baḥran wa-badran ...* (Abd al-Qāhir al-Jurjānī 1991, p. 403)

> ... One can only think rationally about *tashbīh* when there is both a *mushabbah* and a *mushabbah bi-hi*. Now if that is the case when a *tashbīh* is simple and explicit, it is even more important when a *tashbīh* is exaggerated [for literary effect]. [I.e.,] when the *mushabbah* completely transforms into the genus of *the mushabbah bi-hi* and, for example, man becomes lion, sea, or moon...?

In the quoted passage, it appears that ʿAbd al-Qāhir al-Jurjānī uses the term *tashbīh mursal* to refer to simile in a broad sense, encompassing phrases that contain both a tenor and a vehicle. This is in contrast to *istiʿāra* (metaphor), which he describes as having only the vehicle. In a metaphor, the tenor is not posited as similar to the vehicle, as is the case in a simile, but rather it is treated as identical to the vehicle, resulting in a complete substitution of the tenor by the vehicle. That is, the tenor becomes the vehicle itself, or a member of its genus. The man, for example, is not as strong as the lion (as in a simile, "he is like a lion"), or as generous as a sea which shares its water with everybody, or as beautiful as the moon. Instead, he becomes a lion (as in the metaphor in *raʾaytu l-asada*, "I saw the lion", referring to a man of great strength), and he becomes the sea and the moon. Thus ʿAbd al-Qāhir al-Jurjānī uses the term *tashbīh mursal* to refer to all types of simile, whether it contains a simile particle or not, and he uses [*tashbīh*] *ʿalā maʿnā l-mubālagha* ([*tashbīh*], which is exaggerated [for literary effect]) to refer to metaphor.[6] In another part of his book, ʿAbd al-Qāhir al-Jurjānī uses the phrase *al-tashbīh al-ẓāhir wa-l-qawl al-mursal* to describe a simile. This phrase combines the "explicit *tashbīh*" and "loose speech" to characterize the nature of a simile. (Abd al-Qāhir al-Jurjānī 1991, p. 292). This indicates that, for him, *mursal* means the same as *ẓāhir* (explicit). A phrase that contains a tenor and vehicle with or without a simile particle is clearly a simile phrase and is therefore either a *tashbīh ẓāhir* or a *tashbīh mursal*.

The two types of *tashbīh* appear, without being so described, in the famous book *Miftāḥ al-ʿulūm* (*Key to the Sciences*) by the great rhetorician al-Sakkākī (d. 626/1229). He, however, characterises the second type, without the simile particle, as more eloquent (*ablagh*) than the first:

*wa-ʿlam an laysa mina l-wājibi fī l-tashbīhi dhikru kalimati l-tashbīhi, bal idhā qulta "Zaydun asadun" wa-ktafayta bi-dhikri l-ṭarafayni ʿudda tashbīhan mithlahu idhā qultta "ka-anna Zaydan al-asadu" Allāhumma illā fī kawnihi ablagh.* (Al-Sakkākī 1987, p. 352)

Know that it is unnecessary in the *tashbīh* to mention the word of the simile. If you say "Zayd is a lion", while being sufficient in mentioning only the two parts [i.e., the *mushabbah* and *mushabbah bi-hi* without the particle], it would be a simile the same as if you say "Zayd is like a lion", but [the first] is more eloquent.

It is in later works, such as those of al-Khaṭīb al-Qazwīnī (d. 739/1338), that we find the terms *tashbīh mursal* and *tashbīh muʾakkad*, which distinguish between the types of *tashbīh* for similes with and without simile particles. These terms help to categorize and clarify the different forms of similes in Arabic rhetoric:

*wa-bi-ʿtibāri adātihi immā muʾakkadun wa-huwa mā ḥudhifat adātuhu mithlu qawlihi taʿālā "wa-hiya tamurru marra l-saḥābi [al-Naml:88] [...] aw mursalun wa-huwa bi-khilāfihi ka-mā marra.* (Al-Khaṭīb al-Qazwīnī 2010, pp. 96–97)

Considering the particle, the *tashbīh* is either *muʾakkad*—this is when the particle is omitted, such as in the Almighty saying: "[the mountains] pass away [as] the clouds pass away" [Q27:88] [...] or *mursal* and this is the contrary [i.e., does have a particle of simile].

Study of the explicit and implicit interrelationship between existing pre-Islamic poems and the qurʾanic text is vital in comprehending the nature of the two compositions. Previous studies have discussed in detail the direct and indirect reliance of qurʾanic text on biblical and fictional materials accessible in Arabia prior to Islam,[7] contributing to new understanding of some qurʾanic suras. This study takes that research a step further by studying the interrelationship between qurʾanic composition and the Arabic poetry with which Arabs were familiar before and during the Revelation. My objective is similar to that of the earlier studies, with the additional goal of unravelling the way in which each of the two corpora was composed. This is a lengthy and bumpy voyage: the number of poems is large and the fields of comparison diverse. Among them is the structure of the text, its

rhetorical fabric, semantics and more. Each field comprises different subfields, themselves further subdivided. My focus here is a niche related to the larger area of rhetoric.

But before going further into this topic, I should confess to a methodological obstacle that confronts any study of this type. This is the authenticity of pre-Islamic poetry and, to some degree, even that of the standard version of the qurʾanic text, the ʿUthmanic Qurʾan. Some Arabic poetry scholars believe that all poems attributed to the pre-Islamic era are fabrications from later periods. While this, today, seems unlikely,[8] it is undeniable that the versions of some poems known today are the outcome of centuries of sifting, tweaking, suppression, and substitution. This may even be true of the Qurʾan itself. There is scholarly belief that the Qurʾan in its ʿUthmanic form is a collective later work that has rearranged and refined the original text. (See, for example, Claude Gilliot 2008, pp. 88–108). If this is so, both the original texts of pre-Islamic poetry and the Qurʾan are beyond retrieval, and our only choice is to work with those which persist.

Illuminating qurʾanic study with Arabic poetry and vice versa in modern research dates back a century to Joseph Horovitz. (Josef Horovitz [1926] 2013). Several decades later, Toshihiko Izutsu investigated *kufr* (concealment/disbelief) in classical Arabic poetry as a way of understanding it in the qurʾanic text. (Toshihiko Izutsu [1966] 2002). Thomas Bauer examined the grammatical and semiotic usage of the word *kull* (all, each, many) in Arabic poetry to interpret the qurʾanic verses in which it appears. (Thomas Bauer 2010, pp. 699–732). Georges Tamer analyzed *dahr* (time/fate) in the poetry corpus and in the Qurʾan, attributing its use in both to Hellenist influence. (Georges Tamer 2011, pp. 21–41). Angelika Neuwirth examined development of the themes *aṭlāl* (effaced abode of the beloved) and *ʿādhila* (reproaches) from the poetry to the Qurʾan. (Angelika Neuwirth 2016b, pp. 25–55). Ghassan El Masri focused on a pre-Islamic poem by al-Aswad b. Yaʿfur al-Nahshalī (d. ca. 600 CE), showing its parallel content with qurʾanic suras. (Ghassan El Masri 2017, pp. 93–135). Nicolai Sinai similarly compared a narrative about the ancient Thamūd tribe in a poem by Umayya b. Abī l-Ṣalt (a contemporary of the Prophet Muḥammad) with that in **Q**91, concluding that the Qurʾan used pre-qurʾanic material, such as Umayya's poem, which it modified to deliver a divine or prophetic message (Nicolai Sinai 2011, pp. 397–416). In another study, he uses poetry to reconstruct pre-qurʾānic notions of *Allāh* (the God), a word that appears frequently in the Qurʾān. (Nicolai Sinai 2019b). Susanne Stetkevych compared the legend of King Solomon related in a pre-Islamic poem by Al-Nābigha l-Dhubyānī (d. ca. 604 CE) with the same story told in **Q**27:15–44 and **Q**38:30–40. (Suzanne Pinckney Stetkevych 2017, pp. 1–37). Ghassan El Masri surveyed the root ʾ-kh-r to interpret the idea of *ākihra* (last day), a central qurʾanic theme. In the same study, he surveyed words related to the notion of fate in classical Arabic poetry and compared them with those in the Qurʾan. (Ghassan El Masri 2020). Lastly, Simon Loynes compared the semantics of the roots *n-z-l* and *w-ḥ-y* in the Qurʾan and in the poetry corpus. (Simon P. Loynes 2021).

All these studies aimed to shed light on the semantic interrelationship between the Qurʾan and the poetry. Rhetoric, the other main field of comparison and the focus of this article, received far less attention from modern scholars, possibly because there are fewer experts in classical Arabic rhetoric. Noteworthy among the rare studies in this area are those of Muḥammad Abdel Haleem (1992, 2020) and Thomas Hoffmann. Both focus on grammatical elements that contribute to rhetoric, and both, particularly Hoffman, stress understanding qurʾanic verses whose poeticity had not been considered. (Thomas Hoffmann 2004, pp. 35–55; 2006, pp. 39–57; 2007; 2009, pp. 65–76).

The REI identifies some 2967 loose similes in the pre-Islamic corpus and 315 in the Qurʾan. Their division by the total number of verses in both corpora shows use of this type of simile to be far more frequent in the poetry than in the Qurʾan—in fact, three times greater (about 13% compared with 5%).

Figures 1 and 2 present the main simile particles used in each of the two corpora. Figure 3 compares them statistically.

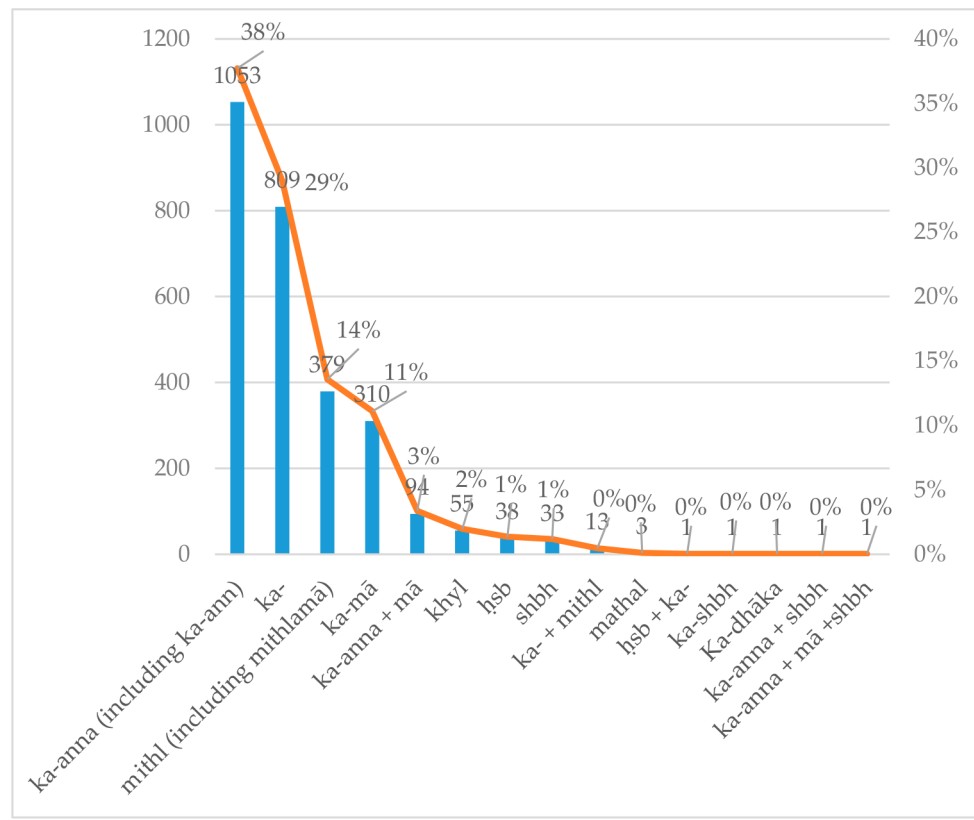

**Figure 1.** Principal simile particles in the poetry corpus.

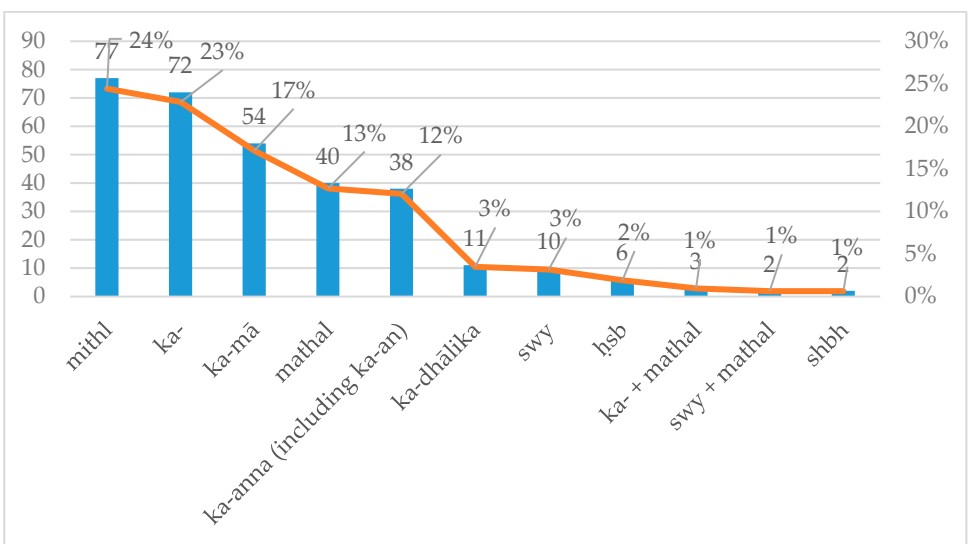

**Figure 2.** Principal simile particles in the Qurʾan.

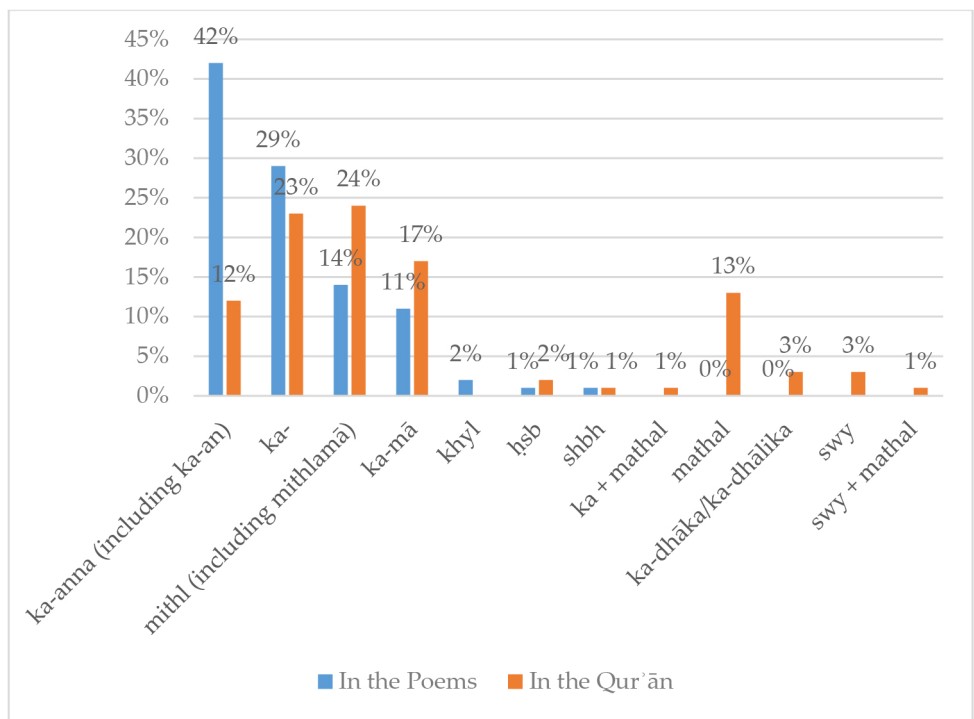

**Figure 3.** Statistical comparison between simile particles used in the two corpora.

The particle *ka-anna* (including *ka-an*, both meaning "as") significantly characterises the poetry, and *mathal* (its example/parable/analogy) is unique to the qurʾanic text. There are, however, five simile particles whose use in the Qurʾan and pre-Islamic poetry is statistically close: (1) *ka-* (as), whose percentage use is slightly greater in the poetry than in the Qurʾan; (2) *ka-mā* (such as), used slightly more often in the Qurʾan than in the poetry; (3) *mithl* (like), again, used slightly more often in the Qurʾan than in poetry; and two infrequently used particles (4) *ḥsb* derivatives (deem) and (5) *shbh* derivatives (looks like, similar to).

These five particles are the focus of this article. It compares their usage in the Qurʾan and the poems and discusses: (1) *Construction*: Primarily the grammatical structure of the sentence in which the simile particle appears (mainly the particle and the vehicle which follows it); (2) *Location*: Where the simile appears. Does it open the verse? Close it? Is it in its centre? Or does it cover the whole verse? (3) *Lexica*: Are there similar words used in these similes in the two corpora? (4) *Type*: In which types of simile is this particle most often used?[9]

The article examines four types or patterns of the *tashbīh mursal*. All were derived from the texts themselves—that is, analysis of examples found in the two corpora underlies their classification into these sub-types:

Type A, the *short simile*, in which the vehicle is either a single word (such as *ka-l-ḥijārati*, like the stones [**Q2**:74]) or an annexation construct (*iḍāfa*) (such as *ka-hayʾati l-ṭayri*, like the figure of a bird [**Q3**:49]).

Type B, the *prolonged simile*, in which the vehicle is prolonged or enriched by a description, whose omission does not alter the essential meaning that the simile conveys. Prolongation can be a single word—such as the adjective in the qurʾanic verse which compares the two halves of the sea split by Moses to *ka-l-ṭawdi **l-ʿaẓīm*** like a **high/big** mountain [**Q26**: 63]. Omitting "high/big" from the simile still conveys that the two halves of the sea resembled a mountain.

Type C, the *analogy based on simile*, in which the vehicle is a complete sentence, deletion of any part of which changes the simile's meaning. An example is comparing the Qurʾan's "goodly word" to *ka-shajaratin ṭayyibatin aṣluhā thābitun wa-farʿuhā fī l-samāʾ*, a goodly tree, its roots set firm, its branches reaching into heaven [**Q14**:24]. "Its branches reaching into heaven" is essential to the simile, indicating that the deeds of believers who speak this

goodly word are counted in the hereafter. This type of simile appears in mediaeval books of Arab rhetoric, where it is named *tamthīl*. ʿAbd al-Qāhir al-Jurjānī defines it, in his discussion of two similarities between the *mushabbah* and *mushabbah bi-hi*. One derives from the *mushabbah bi-hi*, a single word (such as saying "a speech is sweet like the honey": honey is sweet and a speech is metaphorically sweet like the honey). The other, the *tamthīl*, connects with more than one word:

> . . . *ka-qawlihim "huwa ka-l-qābiḍi ʿalā l-māʾ" wa- "l-rāqimi fī l-māʾ," fa-l-shabahu hāhunā muntazaʿun mimmā bayna l-qabḍi wa-l-māʾi, wa-laysa mina l-qabḍi nafsihi, wa-dhālika anna fāʾidata qabḍi l-yadi ʿalā l-shayʾi an yaḥṣula fī-hā, fa-idhā kāna l-shayʾu mimmā lā yatamāsaku, fa-fiʿluka l-qabḍa fī l-yadi laghwun. Wa-kadhālika l-qaṣdu fī "l-raqmi" an yabqā atharun fī l-shayʾi, wa-idhā faʿaltahu fī-mā lā yaqbaluhu, kāna fiʿluka ka-lā fiʿlin . . ."[10]*

> As they say, "he is like one grasping water" and "he is like one writing on the water". The similarity here comes from the relationship between "grasping" and "water", not solely from "grasping". This is because to grasp something with the hand means that it is obtained by it. If it cannot be held, then trying to grasp it would be to no avail. The same can be said about "writing". [Writing] should leave marks on the surface that is written on. If you write on a surface that cannot preserve these marks, then your deed is a non-deed. . .

ʿAbd al-Qāhir al-Jurjānī explains that in the *tamthīl*, the *mushabbah bi-hi* is a construction, and not only a single word. The similarity between the two elements of the *tamthīl*; i.e., the *mushabbah* and *mushabbah bi-hi* is found only when looking at the *mushabbah bi-hi* construction as a whole without dividing it into parts.

Type D, the *compound simile*, comprises two tenors and two vehicles. The tenors produce a single image which is compared with a second image created by the two vehicles. An illustration is the verse which likens the long black hair of the beloved (tenor A) and her white skin (tenor B)—together producing an image of the body of the beloved—to a black cloth (vehicle A) adorned with jewellery (vehicle B)—together creating an image of decorative female clothing (Al-Aʿshā Maymūn 1969, 19:2):

> *Idhā jurridat yawman ḥasibta khamīṣatan/ ʿalay-hā wa-jiryālan yuḍīʾu dulāmiṣā*

> When she [the beloved] is naked, you think as though her body is wearing a black cloth and adorned with shining gold

This type of simile is also recognized by Arabic rhetoricians, who call it *tashbīh murakkab* (a compound simile). It is divided into two main types: *tashbīh mufrad bi-murakkab* (comparing a single [tenor] with a compound [vehicle]) and *tashbīh murakkab bi-murakkab* (comparing a compound [tenor] with a compound [vehicle]). This article focuses on the second type, because in our corpus the first is extremely rare. The renowned rhetorician Saʿd al-Dīn al-Taftāzānī (d. 739/1390) defines this type of simile in his discussion of an example from a verse by Bashshār b. Burd (d. ca. 168/785):[11]

> *wa-l-murakkabu l-ḥissī fī l-tashbīhi lladhī ṭarafāhu murakkabāni, ka-mā fī qawli Bashshār: ka-anna muthāra l-naqʿi, min āthāri l-ghubāri hayyajahu, fawqa ruʾūsina wa-asyāfanā laylun tahāwā kawākibuhu; ay yatasāqaṭu baʿḍuhā ithra baʿḍin . . . fa-wajhu l-shabahi murakkabun ka-mā tarā, wa-kadhā l-ṭarafāni; li-annahu lam yaqṣid tashbīha l-naqʿi bi-l-layli wa-l-suyūfa bi-l-kawākibi, bal ʿamada ilā tashbīhi hayʾati l-suyūfi, wa-qad sullat min aghmādihā, wa-hiya taʿlū wa-tarsubu wa-tajīʾu wa-tadhhabu wa-taḍṭaribu iḍṭirāban shadīdan wa-tataḥarraku bi-surʿatin ilā jihātin mukhtalifatin, wa-ʿalā aḥwālin tanqasimu ilā l-iʿwijāji wa-l-istiqāmati wa-l-irtifāʿi wa-l-inkhifāḍi maʿa l-talāqī wa-l-tadākhuli wa-l-taṣādumi wa-l-talāḥuqi, wa-kadhā fī jānibi l-mushabbahi bi-hi; fa-inna li-l-kawākibi fī tahāwīhā tawāquʿan wa-tadākhulan wa-stiṭālatan li-ashkālihā.*

The compound simile, which is perceivable by sense perception, whose two parts (*mushabbah* and *mushabbah bi-hi*) comprise the compound, resembles a phrase found in Bashshār's verse: "*The raised dust*" (that is, the dust which is raised high)

"*above our heads, and our swords are like a night whose stars fell down*" (that is, they fell one after the other). As is seen, the similarity [itself] is compound, as are the two parts [of the simile]. He [the poet] did not mean to compare the dust with the night, and the swords with the stars; rather, his intention was to describe the situation of the swords once they were unsheathed, and when they were rising and falling, moving rapidly in different directions and different patterns, awry and straight, ascending and descending, meeting and colliding, and following one another. The same [can be said] regarding the *mushabbah bi-hi*: when the stars fall, they drop, clash together and their shapes elongate.

This verse describes an active battlefield. The air is so thick with dust (churned up by the hooves of the galloping horses) that all that is visible is the slashing of the gleaming swords, rising and falling. The poet compares this with stars falling in the black of night. According to Saʿd al-Dīn al-Taftāzānī, the purpose of this verse is to construct a compound image that compares swords flashing in clouds of dust to stars falling in pitch darkness. Both the *mushabbah* and the *mushabbah bi-hi* produce compound images. The similarity between the two is not solely a comparison of the simile's isolated elements. Bashshār was not comparing the dust to the night or the swords to the stars. His comparison was between the verse's compound images.

Each of the five simile particles is discussed individually. The discussion is preceded, (other than for the last two particles—the derivatives of *ḥsb* and *shbh*—whose recurrence is comparatively scarce) by a figure, which presents the principal comparative statistical data related to each simile particle. The figures aim to facilitate the discussion that follows. Whereas they give only percentages of occurrences, the discussion refers, where necessary, to the whole numbers of occurrences.

## 2. *Ka-*

The poems tend to use the *ka-* simile in short similes (52%), neglecting analogy (only about 5% of the total *ka-* similes). This is reversed in the Qurʾan, where the *ka-* simile mostly appears in analogies (51%) its use in short similes dropping to about a third of the *ka-* similes (30%). Figure 4 compares the types of *ka-* similes that appear in the two corpora:

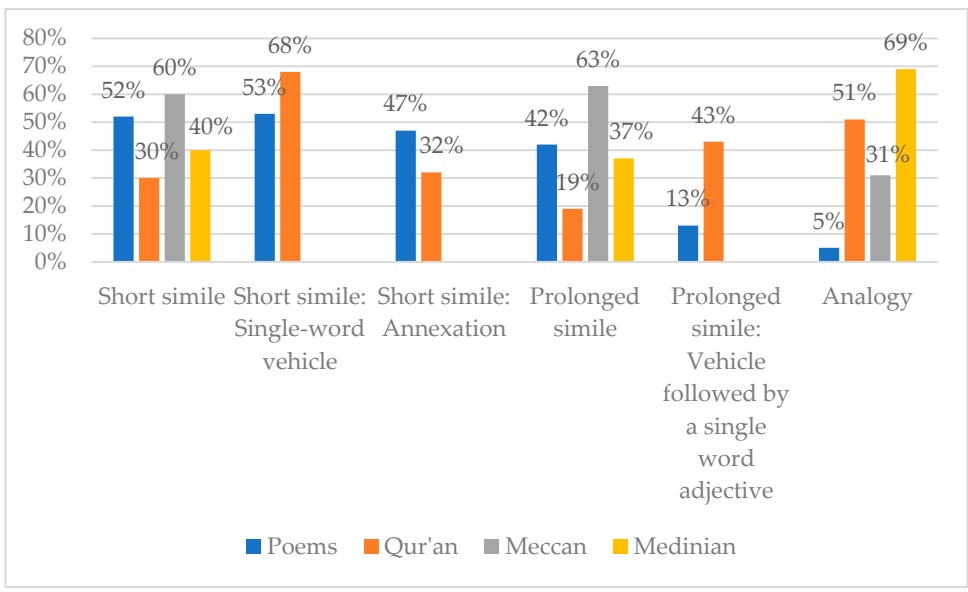

**Figure 4.** Statistical comparison between the *ka-* simile in the two corpora.

In short similes, the poems use each of the vehicle's two constructions almost equally—the single-word vehicle (53%) and annexation (47%). In the Qurʾan, on the other hand, the single-word vehicle is the more prominent (68% compared with about 32% for annexation).

It should be noted that qurʾanic dependence on short similes is greater in the Meccan suras, that is, those of the earlier stages of the Revelation during the Prophet Muḥammad's stay in his hometown of Mecca. Their number decreases in the Medinian suras, revealed after the Prophet moved to Medina in 622 CE.[12] This is opposite to what occurred with the analogy, which flourished during the Medinian period.[13]

Another type of *ka-* simile, the prolonged simile, is also found in the two corpora, though more often in the poetry than in the qurʾanic text (42% vs. 19%, respectively). In the Qurʾan, this type of simile again appears mostly in Meccan suras.[14] The prolonged simile has different constructions, which include six different constructions in the Qurʾan and about 95 in the poems. One of these constructions, common to both the Qurʾan and the poems, is when the vehicle is followed by a single adjective (about 43% in the Qurʾan vs. 13% in the poems). Another, which is not seen in the Qurʾan, appears in 13% of *ka-* similes in the poetry. It comprises a three-word vehicle: the first two words have annexation construction and the third is an adjective—for example, *ka-waqʿi l-mashrafiyyi l-muṣammami* (Aws b. Ḥajar 1979, 48:41) (like the blow of a sharp Mashrafite sword).

In both corpora, the largest number of prolonged similes, in which a single adjective follows the vehicle, appears at the ends of verses. Qurʾanic verses end with: *ka-l-ʿurjūni l-qadīm* (like an old dry palm branch) [**Q36**:39]; *ka-l-ṭawdi l-ʿaẓīm* (like a high/big mountain) [**Q26**:63]; *ka-l-farāshi l-mabthūth* (like scattered moths) [**Q101**:4]; *ka-l-ʿihini l-manfūsh* (like colourful carded wool) [**Q101**:5]; and *ka-ʿaṣfin maʾkūl* (like the chewed-up chaff) [**Q105**:5]. The poetry is awash with examples of such construction. The following, in my view, echo qurʾanic verses (though not necessarily qurʾanic similes), both structurally and in their content and/or phraseology. All except one are from the Meccan suras [**Q63**]:

(1) *ka-l-luʾluʾi l-munkharim* (like pierced pearls) (Al-Aʿshā Maymūn 1969, 4:44),[15] which, I venture, reflects the qurʾanic *ka-amthāl l-luʾluʾi l-maknūn* (like pearls hidden in their shells— that is, unpierced) [**Q56**:23]. (2) *ka-l-shihābi l-mūqad* (like a piercing flame) (Al-Nābigha l-Dhubyānī 1996, 36:10), *ka-l-qabasi l-multahib* (like a burning flame) (ʿAntara b. Shaddād 1992, 4:4), and *ka-l-nujūmi l-thawāqib* (like star of piercing brightness) (ʿAntara b. Shaddād 1992, 4:5)—all of which recall *al-najm l-thāqib* [**Q86**:3] and *shihāb thāqib* (a piercing flame) [**Q37**:10]. (3) *ka-l-ʿāriḍi l-haṭil* (like a rain cloud) (ʿAntara b. Shaddād 1992, 117:7), which shares the meaning of *ʿāriḍun mumṭirunā* (this cloud will bring rain) [**Q46**:24]. (4) *ka-l-jabali l-rāsī* (like a mountain standing firm) (ʿAntara b. Shaddād 1992, 72:7), echoing the aforementioned *ka-l-ṭawdi l-ʿaẓīm* [**Q26**:63]. (5) *ka-l-ʿasībi l-mushadhdhab* (like a palm tree trunk stripped of its spines) [ʿĀmir b. al-Ṭufayl (ʿAbīd b. al-Abraṣ and ʿĀmir b. al-Ṭufayl 1913), 40:5], resonating, to some degree, with *ka-l-ʿurjūni l-qadīm* [**Q36**:39]. (6) *ka-l-khashabi l-shāʾil* (like raised timbers) (Imruʾ al-Qays 2000, 14:8), echoing *ka-annahum khushubun musannada* (as if they were timbers [firmly] propped up) [**Q63**:4]. And (7) *ka-l-farāshi l-mushfatirr* (like scattered moths) (Ṭarafa b. al-ʿAbd 2003, 17:31), which echoes *ka-l-farāshi l-mabthūth* [**Q101**:4].[16]

My point here is not that the qurʾanic verses were directly influenced by the poetry, but that recurrence of such phrases in poetry may indicate certain formulaic sentences or templates common in pre-Islamic poetry, which are represented in the verses quoted above. By "formulaic sentence" or "template", I mean a particular grammatical structure which recurs in different literary texts. This structure sometimes includes recurrent lexical items and is sometimes specifically positioned in the verse.[17] This suggests that a certain expression or structure was encoded in the minds of those who composed the texts. The appearance of the same formulaic sentences/templates in two different texts does not, therefore, necessarily indicate that text B (composed later) was directly influenced by text A (composed earlier), but that both made use of a familiar textual expression. Since the Qurʾan, as is firmly attested by classical scholars,[18] aimed for greater eloquence than the poetry familiar to pre-Islamic Arabians, it is unsurprising that it used poetic formulas.

A construction often used in the Qurʾan is *ka* + a relative pronoun. Rarely, it is *man* (*ka-man*—like that who. . . [**Q47**:15]) and *alladhī* (*ka-lladhī*—like that who. . .) [**Q6**:71; **33**:19]. More frequently it is the plural *alladhīna* (*ka-lladhīna*—like those who. . .—which appears ten

times).[19] Most are in Medinian suras, other than **Q6**:71 and **Q45**:21 which Theodor Nöldeke assumes were composed in the third late Meccan period. (Theodor Nöldeke 2004, pp. 130, 145). *Ka-man* and *ka-llhadhīna* are not found in the pre-Islamic corpus. *Ka-lladhī* appears only four times in the entire corpus.[20] Similarities between phrases that include *ka-lladhī* and those in the Qurʾan were not attested.

The *ka-* also appears with the words *mathal* and *mithl* and their plural form *amthāl*. The first is unique to the Qurʾan and used more in the Medinian suras than the Meccan.[21] It has no parallel use in the poetry. The construction of *ka-* + *mathal* is sometimes followed by one of the two relative pronouns *alladhīna* and *man: ka-mathali lladhīna* (their parable/example is like those who) [**Q59**:15] or *ka-man mathaluhu* (be his parable like the one…) [**Q6**:122]. *Ka-mathal* is seen in two qurʾanic verses. The first compares worldly life with vegetation: *ka-mathali ghaythin aʿjaba l-kuffāra nabātuhu thumma yahīju fa-tarāhu muṣfarran thumma yakūnu ḥuṭāman* (Its parable is that of vegetation that flourishes after rain: the growth of which delights the tillers, then it withers and you see it turn yellow, soon it becomes dry and crumbles away) [**Q57**:20]. The other compares the relationship between hypocrites and their followers: *ka-mathali l-shayṭāni idh qāla li-l-insāni ukfur fa-lammā kafara qāla innī barīʾun min-ka innī akhāfu llāha rabba l-ʿālamīn* (Their parable is like Satan when he says: "Man! Disbelieve!" But when man becomes a disbeliever, he says: "I have nothing to do with you; I fear Allah, the Sustainer of worlds") [**Q59**:16]. All these examples, other than **Q6**:122, are Medinian.

Turning to the *mithl*, there are a few examples in the Qurʾan and the poetry corpus in both its singular and plural constructions—*ka-mithli* and *ka-amthāli*. Similarity between the two corpora in use of the first form is not seen, whereas that of the second form is manifest.[22] In its other form, *ka-amthāli*, it constitutes a complete Meccan verse in the Qurʾan, describing the women in Paradise—*ka-amthāli l-luʾluʾi l-maknūn* (like unto pearls hidden in their shells) [**Q56**:23]. Here, the simile's structure is similar to that seen in the poems. As in this example, the vehicle is followed by a single-word adjective that describes it. The tenor is in a preceding qurʾanic verse—*wa-* + a single noun followed by an adjective, *wa-ḥūrun ʿīn* (and dark-eyed damsels). In the poetry, the *ka-amthāli* simile occurs eight times.[23] The vehicle sometimes has qurʾanic structure and is followed by a single-word adjective or a "state of consciousness" (*ḥāl*) that describes it. This simile by Bishr b. Abī Khāzim (d. 598 CE), which compares ibex horns with wooden howdah poles covered in colourful fabric, is an example: *ka-amthāli l-ʿarīshi l-mudammami* (like the coloured poles of the howdahs) (Bishr b. Abī Khāzim 1994, 40:18). In other instances, the tenor has the qurʾanic structure of *wa-* + the tenor (a single word), which is sometimes, but not always, followed by an adjective. Unlike the qurʾanic verse, however, when the adjective is used in poetry, the vehicle separates it from the tenor. An example of this structure without the adjective is the verse of Al-Aʿshā Maymūn's (d. after 3/625) which describes black-eyed women, not of paradise this time but on earth: *wa-ḥūrun ka-amthāli l-dumā* (and dark-eyed damsels like idols) (Al-Aʿshā Maymūn 1969, 33:11). An example with the adjective from the same poet is: *wa-bīḍun ka-amthāli l-ʿaqīqi ṣawārimun* (and cutting swords like the carnelian) (Al-Aʿshā Maymūn 1969, 30:27).

A rare noun used with *ka-* is *dʾb* (manner/habit/what happened to). Compared with other nouns to which the simile particle is suffixed, its appearance is scarce in both corpora, possibly making it a unique construction, deserving of further illumination. The Qurʾan uses it to describe the punishment inflicted on ancient peoples, comparing it with that suffered by those who disbelieved the Muḥammadian message. The same verse, with slight alteration in the words, is repeated three times in the qurʾanic text (all Medinian suras). (1) *ka-daʾbi āli Firʿawna wa-lladhīna min qablihim kadhdhabū bi-āyātinā* (the like of what happened to Pharaoh's people and those who lived before them who gave the lie to Our messages [**Q3**:11]); (2) *ka-daʾbi āli Firʿawna wa-lladhīna min qablihim kafarū bi-āyāti llāhi* (the like of what happened to Pharaoh's people and those who lived before them who denied the truth of God's messages [**Q8**:52]); (3) *ka-daʾbi āli Firʿawna wa-lladhīna min qablihim khadhdhabū bi-āyāti*

*rabbihim* (the like of what happened to Pharaoh's people and those who lived before them who gave the lie to their Sustainer's message [**Q8**:54]).

The same *ka-daʾbi* construction is found in two pre-Islamic poems. One is by ʿAdī b. Zayd (d. ca. 600 CE), sent to his brother informing him he was in prison and telling him (ʿAdī b. Zayd 1965, 111: 4):[24]

> *fa-lā aʿrifanka ka-daʾbi l-ghulāmi mā lam yajid ʿāriman yaʿtarim*

> You should not handle as a child when he does not find milk to suckle

The other example, taken from the famous *muʿallaqa* of Imruʾ al-Qays (d. ca. 545 CE), is closer in construction and context to the qurʾanic examples. Grammatically, it resembles annexation: *ka-daʾb + muḍāf ilayhi* + conjunction particle + noun. In both the poetry and the qurʾanic verses, memories of the past are invoked. What happened to Pharaoh's people and their forebears in the Qurʾan and what happened to the poet and his former beloved and neighbour in the poem of Imruʾ al-Qays. The latter reads (Imruʾ al-Qays 2000, 1:7):[25]

> *ka-daʾbika min Ummi l-Ḥuwayrithi qablahā / **wa-jāratihā** ummi l-Rabābi bi-Maʾsili*

> **As was your wont** before her with Umm al-Ḥuwayrith **and her neighbour** Umm al-Rabāb at Mount Maʾsil

One final note related to the *ka-* simile is that, in both corpora, the tenor and vehicle are often two different words. In only a handful of examples is the vehicle repeated as the tenor or as a word derived from the same root as the tenor.

### 3. *Ka-mā*

The *ka-mā* particle is often used in verb-similes, when the tenor and mainly the vehicle are verbs or words with the meaning of a verb, such as infinitives. The vehicle is rarely a noun or pronoun following this particle. In such similes, where the vehicle is a noun or a pronoun, there are no observed similarities in its construction between the Qurʾan and the poems.[26] On the other hand, in verb similes in both corpora, most vehicles following the particle are perfect tense verbs (74% perfect vs. 25% imperfect verbs and 1% other in the poems; and 76% perfect vs. 22% imperfect tense and 2% other in the Qurʾan). In Figure 5, the primary data regarding the use of the *ka-mā* simile in the two corpora is presented:

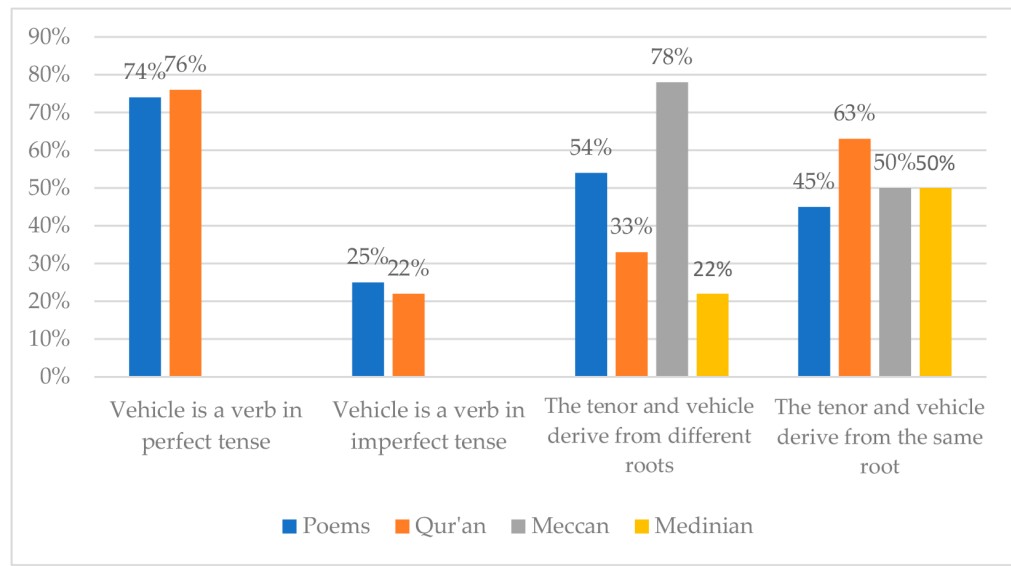

**Figure 5.** Statistical comparison between the *ka-mā* simile in the two corpora.

Most examples are prolonged similes. This is easily explained: the vehicle is often a verb-phrase, requiring a verb followed by additional parts of speech (such as subject and object) and possibly prepositional constructions as well. All this extends the vehicle, making analogies and short similes infrequent.

All short similes in the Qurʾan and the majority in the poems share a consistent structure, with the vehicle including a verb + pronoun: *ka-mā faʿalū*.[27] Of interest are the following examples from the two corpora which, apart from their grammatical structure, share lexica or semantics: ***law kāna*** *maʾa-hu ālihatun* ***ka-mā yaqūlūna*** (If there were other gods besides Him, as they say) [**Q17**:42; Meccan], which has semantic similarity with *fa-**law kāna** ḥaqqan **ka-mā khabbarū*** (if it were true, as they said) [ʿAmr b. Qamīʾa, *Muntahā l-ṭalab* (Ibn Maymūn al-Baghdādī 1999), 15:23], where the verbs *yaqūlūna* and *khabbaru* have similar meanings. This is also true of the verse *wa-**in lam yakun** illā **ka-mā qulti*** (even if it were just as you said) (Aws b. Ḥajar 1979, 48:5); in which the verbs in both the qurʾanic and poetry verses derive from the same root, *qwl*, and share a second root derived from *zʾm* (to claim). The qurʾanic verse contains the phrase *ka-mā zaʾamta* (as you have claimed) [**Q17**:92], while the poetry has phrases such as *ka-mā zaʾamū* (as they claimed) (Al-Aʿshā Maymūn 1969, 2:79), *ka-mā zaʾam* (as he claimed) [Rāshid b. Shihāb al-Yashkurī, *Mufaḍḍaliyyāt* (Al-Mufaḍḍal al-Ḍabbī 1918), 86:2], and *ka-mā kāna zaʾam* (as he has already claimed) (Al-Muthaqqib al-ʿAbdī 1971, 6:22).

In short similes, there is also indirect similarity between the poetry verse *fa-man aṭāʾaka fa-nfaʾhu bi-ṭāʾatihi ka-mā aṭāʾaka* (then whoever obeys you, reward his obedience as he obeyed you) (Al-Nābigha l-Dhubyānī 1996, 1:24) and the qurʾanic *hāʾulāʾi lladhīna aghwaynā aghwaynāhum ka-mā ghawaynā* (these are the people whom we led astray; we led them astray as we were astray ourselves) [**Q28**:63; Meccan]. The first verse is by Al-Nābigha l-Dhubyānī, from a poem which, as shown by Suzanne Stetkevych, has shared contents with qurʾanic stories. (Suzanne Pinckney Stetkevych 2017, pp. 1–37). Both verses use wordplay in which a single root is involved three times. Here, too, it is noteworthy that the examples of *qwl/khbr/zʾm* vehicles all appear in Meccan suras.

As mentioned, the *ka-mā* is mostly used in prolonged similes. These similes are built of one of three main constructs which appear in both the Qurʾan and the poems:

(1) The tenor and vehicle derive from different roots. This structure characterises the poetry, and is the most intensively used, appearing in over half of the corpus's similes (140 of 261 instances, 54%). In the Qurʾan it appears in 18 prolonged similes (33% of total prolonged similes); 14 of them (78%) are in Meccan suras. The construct was abandoned after the migration to Medina.[28] Some examples: *wajnāʾa* ***yaṣrifu*** *nābāhā. . . ka-mā* ***takhammaṭa*** *faḥlu l-ṣirmati. . .* (a strong she-camel; it *grinds* its canine teeth [producing a high sound] like the *roaring* of a stallion) [al-Aswad b. Yaʿfur al-Nahshalī, *Muntahā l-ṭalab* (Ibn Maymūn al-Baghdādī 1999), 52:7]. In the Qurʾan: *ka-mā badaʾakum taʿūdūna* (You shall return to Him as He created you).

(2) The tenor and vehicle derive from the same root. Poetry makes frequent use of this construction (117 of 261 instances, 45%), as in the following verse: *wa-ẓalla hawāki* ***yanmū*** *kulla yawmin / ka-mā* ***yanmū*** *mashībī fī shabābī* (your love continued to *grow* every day; just as my grey hair continued to *grow* within my black hair) (Antara b. Shaddād 1992, 20:2). Its use in the Qurʾan, however, is statistically greater. Most of the Qurʾan's prolonged similes (34 of 54, 63%) have this construction. An example: *kutiba ʿalaykum al-ṣiyāmu ka-mā* ***kutiba*** *ʿalā lladhīna min qablikum* [**Q2**:183] (Fasting is *ordained* for you as it was *ordained* for those before you). Use of this construction typifies the Meccan and Medinian suras alike (50% for each),[29] and therefore seems to be a rhetorical structure that accompanied the Qurʾan from its beginning to the later periods of its Revelation.

(3) In four instances only in the poems and two Medinian instances in the Qurʾan (1% for each), the verse begins with the simile particle, followed by the vehicle. The tenor is mentioned not in this verse but indirectly alluded to in that which precedes it.[30] There are two prominent examples in the poetry. One is by Dhū l-Iṣbaʿ l-ʿAdwānī (d. 600 CA), a pre-Islamic poet whose poetry was rich in gnomics. The verse describes God's power to end the lives of strong people and mighty communities, a familiar motif in the qurʾanic context [Dhū l-Iṣbaʿ l-ʿAdwānī, *Muntahā l-ṭalab* (Ibn Maymūn al-Baghdādī 1999), 122:24]. The other is by ʿAntara b. Shaddād (1992, 82: 3), who also uses *ka-mā* not only as a simile particle but also to mean "because", in the same way it is used in some qurʾanic verses.

Hence, in *ka-mā arsalnā fīkum rasūlan minkum* [**Q2**:151], the verse can translate not only as "*Just as* We bestowed Our favor upon you when We sent among you a messenger of your own", but also as "*Since We* have sent you a messenger . . ."[31] The same semantic function of this *ka-mā* can be applied to these verses by ʿAntara b. Shaddād [d ca. 600 CE]:

> *saqā llāhu ʿammī min yadi l-mawti jarʾatan/wa-shullat yadāhu baʿda qaṭʿi l-aṣābiʿi*

> *ka-mā qāda mithlī bi-l-muḥāli ilā l-radā/wa-ʿallaqa āmālī bi-dhayli l-maṭāmiʿi*

> I hope God will make my uncle drink a cup from the hand of death, paralyse his hands and cut his fingers

> Just as/because he led a person like me to his own death; after he made me following a lying hope

In all types of *ka-mā* simile—short, prolonged, analogy-based—the vehicle can be expressed in dozens of different grammatical structures. These are common to the Qurʾan and the poems:

(1) Verb + subject (*fāʿil*) + prepositional construction (preposition + genitive noun); such as *ka-mā anzalnā ʿalā l-muqtasimīn* (similar to what We sent to those who divided [the Scriptures]) [**Q15**:90, Meccan].[32] And *ka-mā ʿāsha l-dhalīlu bi-ghuṣṣatin* (as the base person spends his life in grief) (ʿAntara b. Shaddād 1992, 23:12).[33] In some instances, the prepositional construction is followed by an object (*mafʿūl bi-hi*), such as *ka-mā arsalnā ilā Firʿawna* **rasūlā** (We sent **an apostle** to Pharaoh) [**Q73**:15; Meccan] vs. *ka-mā ajjajta bi-l-lahabi l-ḍirāmā* (as you make fire fierce by adding **kindling wood**) [ʿĀmir b. al-Ṭufayl (ʿAbīd b. al-Abraṣ and ʿĀmir b. al-Ṭufayl 1913) 2:26].[34]

(2) Verb + subject + object, such as *yaʿrifūnahu ka-mā yaʿrifūna abnāʾahum* (they know this as they know their own sons) [**Q2**:146, Medinian; **6**:26, Meccan]; and *tutʿibu abṭālahā ka-mā atʿaba l-sābiqūna l-kasīra* (it exhausts its warriors just as a victorious horseman in a race exhausts his broken-legged horse [to continue running]) (Al-Aʿshā Maymūn 1969, 12:48).[35] In the next two examples of qurʾanic and poetic verse, each remarkably has two similes: *ka-* followed by the present structure of the *ka-mā*. The qurʾānic example reads *yawma naṭwī l-samāʾa ka-ṭayyi l-sijilli li-l-kutubi, ka-mā badaʾnā awwala khalqin nuʿīduh* (On that day, We shall roll up the heavens like a scroll of writings. As We originated the first creation, so shall We [on that day] produce it again) [**Q21**:104; Meccan]. The other verse describes beautiful women and it reads (Ṭarafa b. al-ʿAbd 2003, 17:25):

> *ka-banāti l-makhri yamʾadna, ka-mā / anbata l-ṣayfu ʿasālīja l-khuḍar*

> They quiver like the white clouds of summer. As green branches of the *khuḍar* trees which grew up in the summer

Note that whereas an equivalent structure, in which the object precedes the subject, appears in the poetry (*ka-mā yahwāhu rumḥī* as my spear loves it ʿAntara b. Shaddād 1992, 39:8),[36] it is virtually absent from the Qurʾan, other than one example from **Q8**:5 (Medinian): *ka-mā akhrajaka rabbuka* (as your Lord brought you out of. . .). This particular structure distinguishes solely the poetry. In it, the vehicle often closes the verse.

(3) Passive verb + *nāʾib fāʿil* (subject of the predicate),[37] sometimes followed, in the poetry,[38] by an adjective (*naʿt*) or in both the poetry and the Quran[39] by a longer, usually prepositional, phrase. In some instances, the vehicle closes the verse in both. In poetry, there are verses which end with phrases such as *ka-mā ḍuriba l-ʿaḍīdu* (as the tree branches are smitten), *ka-mā qtusima l-liḥāmu* (as meat is cut), *ka-mā rtufida l-ḍarīḥu* (as the shrine is raised), and so on; and in the Qurʾan, a verse which ends with *ka-mā ursila l-awwalūna* (like those [prophets] were sent before) [**Q21**:5; Meccan].

### 4. *Mithl*

The salient difference between use of this simile particle in the Qurʾan and in the poetry is that in the latter it appears in both rhetorical (196 of 371 occurrences, 53%) and non-rhetorical similes (175 occurrences, 47%), whereas in the Qurʾan it is restricted to the non-rhetorical. Non-rhetorical similes are those in which the tenor and vehicle are

almost the same (from similar fields, similar contexts) and the simile's aim is to support an argument rather than produce literary images. Examples are: *hali l-mujarribu mithlu man lā yaʿlam* (Is the experienced similar to the one who is not?) (Bishr b. Abī Khāzim 1994, 41:8) and *fa-ʾtū bi-ʿashri suwarin mithlih* (Make up ten suras like this) [**Q11**:13]. The rhetorical simile compares tenors and vehicles from different fields or contexts, producing a certain image or portrait, such as *yukallifūna kulla yaʿmalatin mithla l-mahāti* (they ride on swift she-camels which look like wild cows) [ʿAbīd b. al-Abraṣ (Abīd b. al-Abraṣ and ʿĀmir b. al-Ṭufayl 1913), 25:3]. The image of the wild cow portrays or even replaces that of the she-camel. Figure 6 summarizes the key data related to the *mithl-* simile as compared in the two corpora:

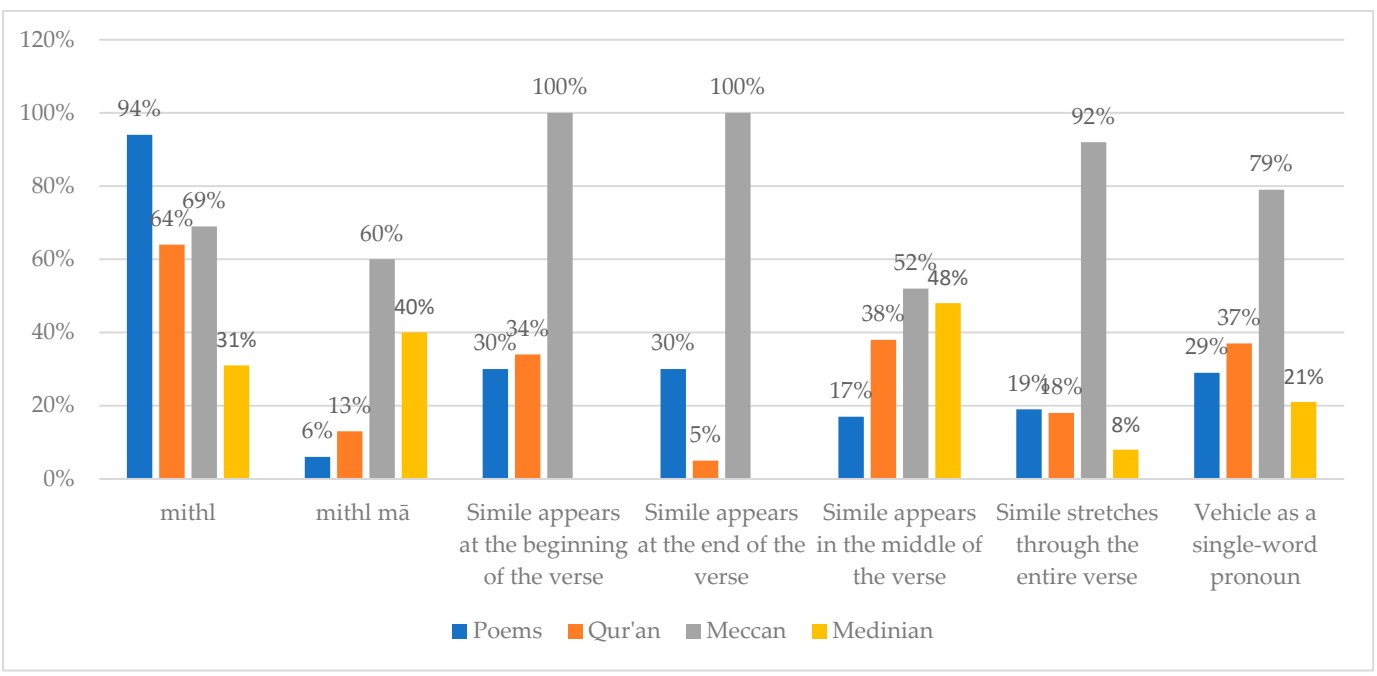

**Figure 6.** Statistical comparison between the *mithl* simile in the two corpora.

Of all 77 *mithl* instances in the Qurʾan, only two (3%) can be considered literary similes. One is *alladhīna yaʾkulūna l-ribā lā yaqūmūna illā ka-mā yaqūmu lladhī yatakhabbaṭuhu l-shayṭānu mina l-mass* (Those who live by usury will not rise up before Allah except like those who are driven to madness by the touch of Satan) [**Q2**:275; Medinian]. The other describes the beautiful maidens of heaven and appears in a Meccan sura [**Q56**:22–23]: *wa-ḥūrun ʿīn, ka-amthāli l-luʾluʾi l-maknūn* (And pure and wide-eyed [females], like unto pearls hidden in their shells). This simile resonates, as shown, with other pre-Islamic similes (such as, Al-Aʿshā Maymūn 1969, 4:44; 33:11). The poetry, on the other hand, describes a judge, likening his face to a shining moon (*fa-qaḍā baynakum ablaju mithlu l-qamari l-bāhiri*) (A judge, [with a face] shining like a white moon, has rendered a judgment between you) (Al-Aʿshā Maymūn 1969, 18:22); and equates the red colour of the earth, soaked in the blood of enemies, to red leather (or red oil, *dihān*): *fa-idhā mā l-arḍu / ṣārat wardatan mithla l-dihān* (and when the earth becomes red like the red leather/red oil) (ʿAntara b. Shaddād 1992, 146:10). Another simile echoes **Q55**:37, a sura that Theodor Nöldeke classified with those of the second Meccan period. (Theodor Nöldeke 2004, pp. 95–96). This sura has a different simile particle, and describes heaven on the Day of Judgement as red leather/red oil—*fa-idhā nshaqqati l-samāʾu fa-kānat wardatan ka-l-dihān* (and when heaven splits asunder and reddens like red oil).

*Mithl* is used 49 times in the Qurʾan (64%), the most frequently used form of this particle. Next is *mithl mā* which is seen here on only 10 occasions (13%). The two forms are often found in the Meccan suras (*mithl*, 69% vs. 31% in Medinian suras; *mithl mā*, 60% vs. 40% in Medinian).[40] Other forms (mainly when the particle is preceded by a preposition)

are used in very low percentages, as shown below. In the poetry, as in the Qurʾan, *mithl* takes the lion's share of these similes, its appearance far more conspicuous than in the Qurʾān, with 290 instances out of 307 (94%). *Mithl mā*, however, is seen in only 6% of instances in poetry. Here too, other forms (mainly those preceded by a preposition) are used. Table 1 presents these minor constructs in the two corpora. Most examples are in Meccan suras:

**Table 1.** Minor constructs of the *mithl* simile in the two corpora.

| Construct | Number of Appearances in the Qurʾan | Number of Appearances in the Poems |
|:---:|:---:|:---:|
| *bi-mithl* | | 18 [1] |
| *bi-mithli mā* | 4 [2] | |
| *bi-mithlihi/bi-mithlihā* | 4 [3] | |
| *fī mithl* | | 1 [4] |
| *ʿalā mithl/ʿalā mithlihi/ʿalā mithlihā/ ʿalā mithlinā/ʿalā mithli mā* | 1 [5] | 11 [6] |
| *ka-mithl/ka-mithlihi/ka-mithlihā* | 1 [7] | 10 [8] |
| *li-mithli hādhā* | 1 [9] | |
| *li-mithlihi / li-mithlihā* | 1 [10] | 4 [11] |
| *min mithlihi/min mithlihimā* | 2 [12] | |
| *mithl alladhī* | 1 [13] | 1 [14] |
| *mithl man* | | 1 [15] |
| *mithlu daʾb* | 1 [16] | |
| *mithlayhim/mithlayhā* | 2 [17] | |
| *amthāl* | 1 [18] | 1 [19] |
| *ka-amthāl* | 1 [20] | 5 [21] |

[1] (ʿAlqama b. ʿAbada 1993), 1:31; 3:13; ʿAlqama b. ʿAbada, *Mufaḍḍaliyyāt* (Al-Mufaḍḍal al-Ḍabbī 1918), 120:16; (Al-Aʿshā Maymūn 1969), 19:17; 23:17; 56:26; 79:10; ʿĀmir b. al-Ṭufayl (ʿAbīd b. al-Abraṣ and ʿĀmir b. al-Ṭufayl 1913), 11:11; 50:1; (Aws b. Ḥajar 1979), 5:3; Ḥājib b. Ḥabīb, *Mufaḍḍaliyyāt* (Al-Mufaḍḍal al-Ḍabbī 1918), 111:3; Al-Ḥārith b. Ḥilliza, *Muntahā l-ṭalab* (Ibn Maymūn al-Baghdādī 1999), 69:68; (Imruʾ al-Qays 2000), 30:5; (Kaʿb b. Mālik al-Anṣārī 1966), 13:6; 32:11; (Al-Mutalammis al-Ḍubaʿī 1970), 5:10; (Ṭufayl al-Ghanawī 1997), 1:65; (Zuhayr b. Abī Sulmā 2004), 48:3. [2]. Meccan: **Q16**:126; **22**:60. Medinian: **Q2**:137, 194. [3]. All Meccan: **Q10**:27; **17**:88 (twice); **18**:109. [4]. Bishr b. Abī Khāzim, *Muntahā l-ṭalab* (Ibn Maymūn al-Baghdādī 1999), 98:3. [5]. **Q46**:10; Meccan. [6]. (Al-Aʿshā Maymūn 1969), 32:35; 65:9; ʿAbdallāh b. Thawr, *Muntahā l-ṭalab* (Ibn Maymūn al-Baghdādī 1999), 467:17; (Aws b. Ḥajar 1979), 14: 11; 35: 14; (Bishr b. Abī Khāzim 1994), 21: 22; al-Find al-Zimmānī, *Muntahā l-ṭalab* (Ibn Maymūn al-Baghdādī 1999), 477: 65; al-Muraqqish al-Aṣghar, *Mufaḍḍaliyyāt* (Al-Mufaḍḍal al-Ḍabbī 1918), 55:14; Qays b. al-Khaṭīm, *Muntahā l-ṭalab* (Ibn Maymūn al-Baghdādī 1999), 343:11; (Al-Shanfarā 1996), 17:51; (Ṭarafa b. al-ʿAbd 2003), 8:40. [7]. **Q42**:11; Meccan. [8]. (Al-Aʿshā Maymūn 1969), 17:18; 22:19; ʿAbīd b. al-Abraṣ (ʿAbīd b. al-Abraṣ and ʿĀmir b. al-Ṭufayl 1913), 30:3; (ʿAmr b. Qamīʾa 1919), 15:10; ʿĀmir. b. al-Ṭufayl (ʿAbīd b. al-Abraṣ and ʿĀmir b. al-Ṭufayl 1913), 7:8; (ʿAntara b. Shaddād 1992), 13:1; 46:19; 64:15; (Aws b. Ḥajar 1979), 14:8; (Imruʾ al-Qays 2000), 45:19. [9]. **Q37**:61; Meccan. [10]. **Q24**:17; Medinian. [11]. (Al-Aʿshā Maymūn 1969), 3:39; 29:21; (Kaʿb b. Mālik al-Anṣārī 1966), 66:16; Suʿdā bint al-Shamardal al-Juhaniyya, *Aṣmaʿiyyāt* (Al-Aṣmaʿī 1993), 27:2. [12]. **Q2**:23 (Medinian); **36**:42 (Meccan). [13]. **Q2**:228 (Medinian). [14]. (Al-Aʿshā Maymūn 1969), 13:19. [15]. (Bishr b. Abī Khāzim 1994), 41:8. [16]. **Q40**:31; Meccan. [17]. **Q3**:13, 165; Medinian. [18]. **Q6**:160; Meccan. [19]. (Kaʿb b. Mālik al-Anṣārī 1966), 26:9. [20]. **Q56**:23; Meccan. [21]. (Al-Aʿshā Maymūn 1969), 33:11; (Bishr b. Abī Khāzim 1994), 40:18; (Laqīṭ b. Yaʿmur 1971), 2:10; (Ṭufayl al-Ghanawī 1997), 2:20; (Zuhayr b. Abī Sulmā 2004), 15:13.

In some instances, phrases, from the two corpora have some similarities. Examples are: (1) the use of *ka-mithl* in these two negative contexts: *laysa ka-mithlihi shayʾun* (There is no one like Him) [**Q42**:11; Meccan] vs. *an lā takūna ka-mithlihi* (that you are not like him) (Al-Aʿshā Maymūn 1969, 17:18). (2) The phrases *an taʿūdū li-mithlihi* (you must [never] repeat something like this) [**Q24**:17; Medinian] and *yasʿā l-ḥalīmu li-mithlihā* (the wise man endeavours to achieve something like it) (Kaʿb b. Mālik al-Anṣārī 1966, 66:16). Here, the verbs which accompany the simile particle *taʿūdū* (to go back) and *yasʿā* (to go toward) have similar literal meanings, although they are given different metaphorical meanings in the

verses ("to repeat" and "to endeavour to achieve"). (3) *wa-la-hunna mithlu lladhī ʿalay-hinna* (they have [rights] similar to those exercised against them) [**Q2**:228; Medinian] vs. *ʿalay-ki mithlu lladhī* (you may receive [benefits] similar to those . . .) (Al-Aʿshā Maymūn 1969, 13:12). Here the prepositional construction, followed by the phrase *mithlu lladhī* (similar to those), appears in the two phrases.[41] (4) *wa-ḥūrun ʿīn ka-amthāli l-luʾluʾi l-maknūn* (and dark and wide-eyed damsels like unto pearls hidden in their shells) [**Q56**:22–23; Meccan] vs. *wa-ḥūrun ka-amthāli l-dumā* (and dark-eyed damsels like idols) (Al-Aʿshā Maymūn 1969, 33:11), *ka-amthāli l-ʿarīshi l-mudammami* (like the coloured poles of the howdahs) (Bishr b. Abī Khāzim 1994, 40:18)—a similarity discussed above.

In the poetry corpus, this simile often appears at the beginning or end of the verse (30% for each). In 19%, it stretches through the entire verse, and in 17% it is in the middle of the verse. In a handful of instances, there are two or three *mithl* similes in the same verse (3%). The Qurʾan makes use of some of the features that characterise the poems and abandons others. The *mithl* simile at the beginning of the verse (34%) is one which it embraces, along with more frequent use of it in the verse's middle. All such similes at the beginning of verses are Meccan,[42] with those in the middle found in Meccan and Medinian suras alike (38%; there are 15 similes in Meccan suras and 14 in Medinian).[43] Similes stretched through the whole verse appear in similar percentages in the poems and the Qurʾan (18%), mainly the Meccan suras (92%), with only one Medinian example [8%, **Q24**:17].[44] Contrary to the poetry, simile at the end of the verse is scarcely used in the Qurʾan (5%),[45] and rarely, as in the poetry, a verse has two *mithl* similes (5%).[46] That is, during the Meccan Revelation, the Qurʾan located the *mithl* simile similarly to the poems, usually at the beginning of the verse, and not infrequently through the whole verse. Again, as in the poetry, the mid-verse simile was used frequently during the Meccan period and after the migration to Medina, while use of the other two locations (at the beginning of the verse or through the whole verse) were almost totally abandoned. Closing with the *mithl* simile was characteristic of the poems only, and greatly neglected in the Qurʾan in both periods.

The most common structure of the simile particle in both the Qurʾan and the poetry is the vehicle as a single-word pronoun, such as A is *mithluhu/mithluhā*. This is found in about a third of the poetry similes (29%)[47] and slightly over a third in the Qurʾan (37%), mostly in the Meccan suras (79% vs. 21% in Medinian suras).[48]

Other than this, the two corpora use different clusters of structures. Most characteristic of the Qurʾan (18%[49] vs. 0.6% in the poems[50]) is the vehicle, following the particle *mithl mā*, as a verbal phrase. Examples are *sa-unzilu mithla mā anzala llāhu* (I can reveal the like of what Allah has revealed) [**Q6**:93] and this verse by ʿAdī b. Zayd (d. ca. 600 CE) (ʿAdī b. Zayd 1965, 103:5):

> *wa-basaṭa l-arḍa basṭan thumma qaddarahā / taḥta l-samāʾi sawāʾan mithlamā faʿalā*
>
> And He spread the earth out, then determined it
>
> Under the sky, adjusted [it] as he made [it].[51]

The subject of this poem by ʿAdī b. Zayd is the creation of the world, and it embraces biblical contents. Kirill Dmitriev has studied in depth the relationships between this poem and the Bible and between some of its verses and the Qurʾan. He contends that the Qurʾan addresses the notion of "spreading the earth" beneath the firmament, but it uses different lexica—mainly derived from the root *frsh*, meaning "plain/plain land" and "to level" [**Q2**:22; **51**:48]. He also uses the root *bsṭ*, which is found in the Qurʾan in other contexts. (Dmitriev 2009, pp. 358–59). It should be noted that ʿAdī's combination of earth and sky in the same verse, as well as that of *basṭ* and the *qadr* (to creed), is occasionally repeated in the Qurʾan in other contexts, such as **Q13**:26; **23**:18; **25**:2; **28**:82; **29**:62; **30**:37; **34**:39; **39**:52, 67; **41**:9–10; **42**:12, 27; **43**:11; **54**:12.

In both corpora, there are instances where the vehicle is followed by a phrase that is part of the tenor. It appears either as (a) an adjective which describes the tenor, following the vehicle; or (b) the vehicle is followed by a prepositional phrase which is part of the tenor. The first characterises the poetry as it appears only once in the Qurʾan: *fa-ʾtū bi-ʿashri suwarin*

*mithlihā muftarayātin* (produce *ten suras* like this, *which are fabricated*) [**Q11**:13; Meccan].
The adjective *muftariyātin* (fabricated) describes the tenor *suwarin* (verses). In poetry, it is
found in few verses (0.3%). An example is *ḥarfin mithli l-mahāti dhaqūni* (a fleet she-camel,
that looks like a wild cow, that relaxes its chin while running) [al-Muraqqish al-Akbar
(Al-Muraqqishān 1998), 17:4].[52] The second characterises the Qurʾan (12%)[53] but is rarely
seen in the poems.[54]

The poetry makes frequent use of a vehicle which is either a single noun (17%)[55] or a
combination of either two nouns or a noun and a pronoun, blended together and related to
annexation. It may be termed a "double annexation". Example: *mithlu ẓahri l-tursi* (like the
back of a shield) (Al-Aʿshā Maymūn 1969, 6:31) (15%).[56] Such vehicles appear only rarely in
the Qurʾan.[57]

**5. Ḥsb and Shbh**

Derivatives of the root 'ḥsb', particularly in the Qurʾan, appear in conjunctive sentences
in both corpora. Examples are ***wa-taḥsabuhum ayqāẓan wa-hum ruqūd*** (**and** you think them
awake, though they were asleep) [**Q18**:18] and ***wa-taḥsibu āyātihinna raqqan muḥīlā*** (Zuhayr
b. Abī Sulmā 2004, 11:2) (**and** you think their remnants as though they were an altered
parchment).[58]

In the poetry, this simile appears not only in conjunctive phrases but also in conditional
contexts (such as *idhā mā mashaw fī l-sābighāti ḥasibtahum suyūlan* (ʿAntara b. Shaddād 1992,
30:11)) (if they walk with their long chainmail coats, you deem they were a flood)[59] and
interrogative phrases (*a-fa-athlan taḥsibuhum*? [Āmir b. Juwayn, *Muntahā l-ṭalab* (Ibn
Maymūn al-Baghdādī 1999), 485:20) (Do you think they are tamarisks?).[60] The conditional
context with the particle *ḥsb* is used only once in the Qurʾan [**Q27**:44]: *fa-lammā raʾathu
ḥasibathu lujjatan* (When she saw it (the floor), she deemed it a pool of water).[61] The
interrogative use is totally absent.

In both corpora, the *ḥsb* derivatives often appear in short similes, such as *fa-taḥsibuhū
īwānā* (Al-Aʿshā Maymūn 1969, 27:6) (you think of [the army] as though it were an arch)
and *ḥasibathu lujjatan* [**Q27**:44].[62] In the poetry, the prolonged simile with the derivatives
of *ḥsb* is common, though less so than the short simile. In the Qurʾan, it is used only once
[**Q76**:19]: *ḥasibtahum luʾluʾan manthūrā* (when you see them, they would seem like sprinkled
pearls). This qurʾanic prolongation of the vehicle by adding a single-word adjective is found
in the poetry in only a single instance: the verse by Zuhayr[63] already quoted. In most cases,
prolongation is a verbal phrase following the vehicle. Occasionally in the two corpora, the
simile that includes *ḥsb* derivatives is an analogy,[64] and in the poetry only, a compound
simile.[65]

In the poetry, most similes include an imperfect verb derived from the root *ḥsb* +
you (masculine) (*taḥsabu/taḥsibu*; 17 instances). Next in frequency is the imperfect + "he"
(*yaḥsabu*; five instances). These are also the most frequent examples in the Qurʾan (twice for
each).[66]

The last particle, derivatives of the root *shbh*, is used only twice in the Qurʾan, in two
forms—*shubbiha* [**Q4**:157] (it is as if it had been so) and *tashābaha* (are alike) [**Q2**:118]. The
latter disappears from the poetry, but *shbh*'s passive form, as in the first qurʾanic example,
recurs.[67]

**6. Summary and Conclusions**

To summarise the main features of the use of simile particles:

*The ka- simile*: (1) In poetry, most *ka-* similes are short, with its two forms (single-word
vehicle and annexation) used almost equally. In the Qurʾan, short similes with this particle
gives way to increased use of analogy—a result of decrease in the use of the short simile
from the Meccan to Medinian periods, and an increase in that of the analogy. In the Qurʾan's
short similes, the single-word vehicle is far more frequent than annexation. (2) The *ka-*
particle is used for prolonged as well as with short similes. This, too, is more common

in the poems than in the Qurʾan. Prolongation in the two corpora is largely similar: a single-word adjective follows the vehicle, and the simile in these instances often closes the verse. (3) The *ka-* particle sometimes prefixes words such as *mathal* (analogy/parable), *mithl* (like) and their plural form *amthāl*. The first is unique to the Qurʾan and is not seen in the poems. It is used significantly more in the Medinian suras than the Meccan. *Ka-* also, but rarely, prefixes the word *dʾb* (and the like of what happened to). In both corpora, *ka-daʾb* is sometimes used for past memories. (4) The poems and the Qurʾan also share the verbal derivation of tenors from roots, which differ from those used for vehicles. This covers the lion's share of similes in the two corpora.

*The ka-mā simile*: (1) This particle is often used in verb-similes. Most vehicles in the two corpora are verbs in the perfect tense. (2) Most similes in the two corpora are prolonged. (3) Their few short similes share the same structure of *ka-mā faʿalū*. (4) In more than half of *ka-mā* similes in the poems, the tenor and vehicle derive from different roots; in less than half, the tenor is a verb similar to the vehicle. In the Qurʾan, this second feature is prominent in both Meccan and Medinian suras. (5) The ways in which the two corpora prolong similes are diverse, but share a few common features. These features are when the vehicle is a verb followed by a subject and prepositional phrase, or by a subject and object. In the poetry only, the object sometimes precedes the subject. Another common feature is when the vehicle is a passive verb followed by *nāʾib fāʿil* (subject of the predicate). *Nāʾib fāʿil* in the two corpora is sometimes followed by a prepositional phrase. Found in the poems but not the Qurʾan is a single-word adjective following *nāʾib fāʿil*.

*The mithl simile*: (1) In the Qurʾan, this particle is used in non-rhetorical similes only, whereas in poetry it serves both rhetorical and non-rhetorical functions. (2) *Mithl* alone is the most common form of this particle in the two corpora, particularly in the poems. It appears more often in the Qurʾan's Meccan suras than its Medinian. Its second most common form in the two corpora is *mithl mā*, which is more characteristic of the Qurʾan than the poems, and more typical of Meccan than Medinian suras. (3) The Qurʾan makes frequent use of certain location features while abandoning others. During the Meccan Revelation, its location *mithl* simile was similar to that in the poems—generally at the beginning of the verse, sometimes stretched through the whole verse, and appearing less often in the middle of the verse. The Medinian suras abandon the first two locations, using only that in the verse's middle. The poems continue to place the simile at the verse's end. (4) The structure most used in both the poems and the Qurʾan (mainly the Meccan suras) is the *mithluhu/mithluhā*.

*Similes that include derivatives of the roots ḥsb and shbh*: (1) In both the Qurʾan (mostly the Meccan suras) and the poems, the *ḥsb derivatives are* often used in conjunctive sentences. In the poetry they also appear in conditional and interrogative phrases. (2) This simile is often found in short similes in the two corpora. It appears in prolonged similes in the poems but rarely in the Qurʾan. Its single prolonged simile in the Qurʾan shares a prolongation not widely used in the poetry: the vehicle is followed by a single-word adjective. (3) The simile that includes *shbh derivatives* is the rarest in the two corpora.

Three main conclusions can be derived from this study. (1) The Qurʾan, in its Meccan period, shares common compositional features with pre- and co-existing Arabic poetry. (2) These features relate to the grammatical structure of the particle phrases, the lexica used in these phrases, and, rarely, the simile's context. (3) During the Medinian period, these features fade away: they are either used far less frequently or fall into total disuse, while other features, absent or rare in the poems, emerge.

Given these correlations, we can postulate the following historical progression: In its earlier stages, the Qurʾan used short and prolonged similes more than analogies, as in the poetry, and it used also specific simile structures which appeared frequently in poetry. There also several qurʾanic verses, mostly Meccan, that share some common lexica with pre-Islamic verses, which may indicate that a more significant adaptation of specific rhetorical models and constructions from poetry occurred during the earlier stages of the Revelation. After the migration to Medina, the Qurʾan deviated from the poetry as a

prototype. It adapted new rhetorical models found less frequently in the poetry—mainly the analogy based on simile, and some constructs such as *ka-man*, *ka-lladhī*, *ka-llhadhīna*, and *ka-mathal*. This may be either because the new contents of the Medinian Qurʾān could be better expressed through these new constructs (Theodor Nöldeke 2004, pp. 153–54), or because these were types and simile constructs used in texts that flourished around Medina.

Investigating the sources of these "Medinian" features is beyond the scope of this article, but a study which compares them with those in other texts known during the Medinian period (such as the Bible) may be useful. Should such studies prove that Medinian similes are closer in structure and possibly in lexica to biblical texts, this would demonstrate that the qurʾanic text followed two proto-compositions: one purely Arabic (Arabic poetry, its beginnings) and one non-Arabic (possibly biblical, maybe translated into Arabic) in its later Revelations. These two proto-compositions are thus important for new understanding of and new insights into the Arabic scripture.

A final observation from this study is that a very large number of the verses which share common features with qurʾanic similes are composed by two pre-Islamic poets, Al-Aʿshā Maymūn and ʿAntara b. Shaddād. (For brevity, not all verses used in this study are referenced.) Future studies comparing pre-Islamic poetry and the Qurʾān should, therefore, pay special attention to the work of these two poets.

Table 2 lists the poets referred to in this article, whose similes share features with those of the Qurʾan.

**Table 2.** Poets and their verses that include features common to qurʾanic similes.

| Poet | Number of Verses |
|---|---|
| Al-Aʿshā Maymūn | 81 |
| ʿAntara b. Shaddād | 51 |
| Imruʾ al-Qays | 24 |
| ʿAdī b. Zayd | 21 |
| Zuhayr b. Abī Sulmā | 20 |
| Aws b. Ḥajar | 18 |
| Bishr b. Abī Khāzim | 13 |
| Kaʿb b. Mālik al-Anṣārī | 11 |
| Ṭarafa b. al-ʿAbd | 11 |
| ʿAbīd b. al-Abraṣ | 10 |
| ʿĀmir b. al-Ṭufayl | 10 |
| Al-Find al-Zimmānī | 7 |
| Al-Nābigha l-Dhubyānī | 7 |
| Taʾabbaṭa Sharran | 7 |
| Ṭufayl al-Ghanawī | 7 |
| Al-Muthaqqib al-ʿAbdī | 5 |
| Al-Aswad b. Yaʿfur al-Nahshalī | 4 |
| Al-Ḥārith b. Ḥilliza | 4 |
| ʿAmr b. Qamīʾa | 4 |
| Qays b. al-Khaṭīm | 4 |

**Table 2.** *Cont.*

| Poet | Number of Verses |
|---|---|
| Al-Akhnas b. Shihāb al-Taghlibī | 3 |
| Al-Muraqqish al-Aṣghar | 3 |
| ʿAlqama b. ʿAbada | 3 |
| Al-Shanfarā | 3 |
| Muʿaqqir b. Ḥimār al-Bāriqī | 3 |
| Al-Ḥārith b. Ẓālim | 2 |
| ʿĀmir b. Juwayn | 2 |
| ʿAwf b. ʿAṭiyya | 2 |
| Dawsar b. Dhuhayl al-Qurayʿī | 2 |
| Ḥājib b. Ḥabīb al-Asadī | 2 |
| Ḥājiz b. ʿAwf al-Asadī | 2 |
| ʿIlbāʾ b. Arqam | 2 |
| Imruʾ al-Qays b. Jabala | 2 |
| Khidāsh b. Zuhayr | 2 |
| Laqīṭ b. Yaʿmur | 2 |
| Muʿāwiya b. Mālik b. Jaʿfar | 2 |
| Rāshid b. Shihāb al-Yashkurī | 2 |
| ʿAbdallāh b. Thawr | 1 |
| ʿAbīd b. ʿAbd al-ʿUzzā | 1 |
| Abu Duʾād al-Iyādī | 1 |
| Abū Qays Ṣayfī b. al-Aslat | 1 |
| Al-Ḥārith b. Waʿla l-Jarmī | 1 |
| Al-Ḥuṣayn b. al-Ḥumām al-Murrī | 1 |
| Al-Jumayḫ al-Asadī | 1 |
| Al-Mufaḍḍal al-Nukrī | 1 |
| Al-Mutalammis al-Ḍubaʿī | 1 |
| Al-Samawʾal b. ʿĀdiyāʾ | 1 |
| ʿĀmir al-Muḥāribī | 1 |
| ʿAmr b. Quʿās al-Murādī | 1 |
| Aʿshā Bāhila | 1 |
| Bayhas b. ʿAbd al-Ḥārith | 1 |
| Bishr b. ʿAmr b. Marthad | 1 |
| Ḍamra b. Ḍamra l-Nahshalī | 1 |
| Dhū l-Iṣbaʿ l-ʿAdwānī | 1 |
| Kaʿb b. Saʿd al-Ghanawī | 1 |
| Maqqās al-ʿĀʾidhī | 1 |
| Muʿādh b. Muʿāwiya b. Jaʿfar | 1 |
| Muraqqish al-Akbar | 1 |

**Table 2.** *Cont.*

| Poet | Number of Verses |
| --- | --- |
| Salama b. al-Khurshub al-Anmārī | 1 |
| Saʿya b. al-ʿUrayḍ al-Yahūdī | 1 |
| Suʿdā bint al-Shamardal al-Juhaniyya | 1 |
| Ṣuḥayr b. ʿUmayr | 1 |
| ʿUbayd b. ʿAbd al-ʿUzzā | 1 |
| ʿUrwa b. al-Ward | 1 |
| Yazīd b. Khaddhāq al-Shannī | 1 |
| Zuhayr b. Masʿūd | 1 |

**Funding:** This research received no external funding.

**Data Availability Statement:** I used the data found in the https://arabic-rhetoric.haifa.ac.il (accessed on 1 May 2023).

**Conflicts of Interest:** The author declares no conflict of interest.

## Notes

[1] https://arabic-rhetoric.haifa.ac.il/ (accessed on 1 May 2023). About the database and its development, see Abd Alhadi et al. (2023). This article outlines how the similes were located in the corpus of texts, as well as describing the generation of the corpus itself. The REI will not be publicly accessible until 2024.

[2] The translation of the term as "unrestricted simile" is taken from Hussein Abdul-Raof (2006).

[3] See, for example, the informative details and different sub-types of simile found in Yūsuf b. Abī Bakr Al-Sakkākī (1987, pp. 332–55). For a comprehensive analysis of al-Sakkākī's contributions to understanding and evaluating *tashbīh*, please refer to William Smyth (1992, pp. 215–29).

[4] Abū l-Ḥasan ʿAlī b. ʿĪsā al-Rummānī, "al-Nukat fī iʿjāz al-Qurʾān", in Khalafallāh and Salām (1968, pp. 80–81).

[5] Abū Hilāl al-ʿAskarī (1952, p. 239). A detailed discussion of the two types of simile, without mention of the terminology that refers to them, appears in other early books on rhetoric. See, for example, the fourth/tenth century work by Al-Qāḍī l-Jurjānī (1966, pp. 442–43). See also Ibn Sinān al-Khafājī (1982, pp. 246–56). Other scholars, such as Ibn Rashīq al-Qayrawānī (d.456/1063-4), distinguish between the two types of simile simply as *tashbīh bi-kāf* (*tashbīh* with [the particle] *ka-*) and *tashbīh bi-ghayri kāf* (*tashbīh* without [the particle] *ka-*) or *tashbīh bi-isqāṭi l-kāf* (*tashbīh* by omitting the *ka-*). See Ibn Rashīq al-Qayrawānī (1981, pp. 293–94).

[6] Elsewhere, it is called *al-tashbīh ʿalā ḥadd al-mubālagha* (*tashbīh* having the sense of exaggeration), see pp. 410, 412.

[7] See, for example, Neuwirth (1981; 2010, pp. 733–78; 2016a, pp. 253–57); Nicolai Sinai (2019a, pp. 215–35; 2017, pp. 219–66).

[8] Opinions about this are detailed in Gregor Schoeler (2006a, pp. 87–110; 2006b).

[9] I depend mainly on the types mentioned in Ali Ahmad Hussein (2015, pp. 47–49).

[10] ʿAbd al-Qāhir al-Jurjānī (1991, p. 104). The author explains this paragraph in detail on pp. 105–9 and cites the term *tamthīl* on p. 108.

[11] Saʿd al-Dīn al-Taftāzānī (2010, pp. 41–42). See the term *tashbīh murakkab* in Fakhr al-Dīn al-Rāzī (2004, p. 131); and the term *al-murakkab bi-l-murakkab* in Al-Sakkākī (1987, p. 338).

[12] For the Qurʾan, I accept the traditional dating which has been independently confirmed by Sadeghi and Bergmann (2010, pp. 343–436). I accept also the Qurʾan's traditional division into Meccan and Medinian suras. Sadeghi summarises the state of the field with respect to the ordering of verses in the Qurʾan, see Behnam Sadeghi (2011, pp. 210–99). Some 60% of the short similes appear in Meccan suras and 40% in Medinian. Meccan suras: **Q7**:179; **11**:42; **16**:77; **25**:24; **31**:32; **34**:13; **42**:32; **44**:45,46; **51**:42; **54**:50; **68**:20, 35; **70**:9; **77**:32. Medinian suras: **Q2**:74; **3**:49; **4**:77, 129; **5**:110; **13**:16; **55**:14, 24, 37; **57**:21. See the division of the Qurʾan into Meccan and Medinian suras in Theodor Nöldeke (2004, 1919); and Nicolai Sinai (2010, pp. 407–39); Nora K. Schmid (2010, pp. 441–60). Unfortunately, due to the large number of similes in the poems, it is not possible to present them here.

[13] In total, 31% of the analogies are Meccan and 69% are Medinian. Meccan suras: **Q6**:71, 122; **11**:24; **14**:18, 24; **16**:92; **21**:104; **42**:11; **45**:21; **56**:16; **57**:20; **59**:15, 16. Medinian suras: **Q2**:17, 19, 171, 259, 261, 264 (twice), 265; **3**:11, 59, 105, 117, 156; **7**:176; **8**:21, 47, 52, 54; **9**:69 (twice); **29**:41; **33**:18, 69; **24**:39; **47**:15; **57**:16; **59**:19; **62**:5; **65**:8.

[14] In total, 63% in the Meccan suras which are **Q26**:63; **29**:10; **30**:28; **31**:28; **36**:39; **38**:28 (twice); **54**:31; **56**:23; **101**:4, 5; **105**:5. The rest are Medinian: **Q2**:200; **24**:63; **33**:32; **49**:2.

15 The large number of poems to which this article refers are marked with the same numbers they have in the REI. Thus, (Al-Aʿshā Maymūn 1969, 4: 44) for example, refers to poem 4 in the REI database, verse 44, composed by Al-Aʿshā Maymūn, Often, these numbers are the same as in certain printed versions of the *dīwān* (poetry collection), where these versions number their poems. Bibliographical details of the printed *dīwān*s from which these poems are taken are given in the References to this article.

16 See ʿAbīd b. al-Abraṣ (Abīd b. al-Abraṣ and ʿĀmir b. al-Ṭufayl 1913), 12:5; 13: 14, 18;30: 9; ʿAbīd b. al-Abraṣ, *Muntahā l- ṭalab* (Ibn Maymūn al-Baghdādī 1999), 74:18; 77:5; 83:9; ʿĀmir b. al-Ṭufayl (Abīd b. al-Abraṣ and ʿĀmir b. al-Ṭufayl 1913), 40:1, 5, 6; (ʿAntara b. Shaddād 1992), 4:4; 64:20; 33:5; 72:7; 117:7, 12; (Al-Aʿshā Maymūn 1969), 2: 60;4: 44; 18:51; 28:11; al-Aswad b. Yaʿfur al-Nahshalī, *Muntahā l-ṭalab* (Ibn Maymūn al-Baghdādī 1999), 55:22; (Bishr b. Abī Khāzim 1994), 23:9; (Imruʾ al-Qays 2000), 1:49; 14:8; 74:21; Khidāsh b. Zuhayr, *Muntahā l-Ṭalab* (Ibn Maymūn al-Baghdādī 1999), 465:9; al-Muraqqish al-Aṣghar, *Mufaḍḍaliyyāt* (Al-Mufaḍḍal al-Ḍabbī 1918), 57:13; al-Muthaqqib al-ʿAbdī, *Mufaḍḍaliyyāt* (Al-Mufaḍḍal al-Ḍabbī 1918), 77:7; (Al-Nābigha l-Dhubyānī 1996), 8:10; 36: 10; 44:24; Qays b. al-Khaṭīm, *Muntahā l-ṭalab* (Ibn Maymūn al-Baghdādī 1999), 346:9; (Al-Shanfarā 1996), 6:4; (Taʾabbaṭa Sharran 1996), 40:2; (Ṭarafa b. al-ʿAbd 2003), 17:31; ʿUbayd b. ʿAbd al-ʿUzzā, *Muntahā l-Ṭalab* (Ibn Maymūn al-Baghdādī 1999), 452:35; ʿUrwa b. al-Ward, *Aṣmaʿiyyāt* (Al-Aṣmaʾī 1993), 10:15, 17.

17 "Formula" is used here slightly differently from its original use. The term is taken from a theory, first developed by Milman Parry (1902–1935) and later used by Albert Lord (1912–1991), for examining epic Greek poetry. About this theory, see M. W. M. Pope (1963, pp. 1–22). Several modern studies have shown that classical Arabic poetry relies heavily on formulas repeated from poem to poem. See James T. Monroe (1972, pp. 1–53); Michael Zwettler (1978); Thomas Bauer (1993, pp. 117–38); Werner Diem (2010, pp. 158–77).

18 This thesis is widely supported, mainly in ʿAbd al-Qāhir al-Jurjānī's two books ʿAbd al-Qāhir al-Jurjānī (1991), and ʿAbd al-Qāhir al-Jurjānī (1992).

19 **Q3**: 105, 156; **8**:21, 47; **9**:69 (twice); **33**:69; **45**:21; **57**:16; **59**:19. I have removed *ka-mā* from this group, since it is used frequently in the Qurʾan; it is considered here as an independent simile particle.

20 ʿAbīd b. al-Abraṣ (Abīd b. al-Abraṣ and ʿĀmir b. al-Ṭufayl 1913), 30: 36; ʿĀmir b. Juwayn, 485, *Muntahā l-ṭalab* (Ibn Maymūn al-Baghdādī 1999): 6; al-Find al-Zimmānī *Muntahā l-ṭalab* (Ibn Maymūn al-Baghdādī 1999), 478: 7, 9.

21 Meccan suras: **Q42**:11; **57**:20; **59**:15, 16. Medinian suras: **Q2**: 17, 171, 261, 264, 265; **Q3**:59, 117; **7**:176; **29**:41; **62**:5.

22 In the Qurʾan, *ka-mithl* is used just once, in a Meccan sura, to describe the inimitability of God: *laysa ka-mithlihi shayʾun* (there is nothing/nobody like Him). It appears at the end of the verse [**Q42**:11], almost closing it. In the poetry, this form appears three of four times toward the end of verses, as in the Qurʾan, but its structure differs. One example from the poetry is *ka-mithli nārin fī yafāʿ* (like fire burning on the heights) (ʿAntara b. Shaddād 1992, 81:11). It describes the spear, but grammatically and structurally this simile differs from the qurʾanic verse quoted above. The other two examples are found in (ʿAntara b. Shaddād 1992, 46:19; ʿAbīd b. al-Abraṣ (Abīd b. al-Abraṣ and ʿĀmir b. al-Ṭufayl 1913), 30:3; and (Imruʾ al-Qays 2000), 45:19.

23 (Al-Aʿshā Maymūn 1969), 30:27; 33:11; (Aws b. Ḥajar 1979), 21:20; 89:20; (Bishr b. Abī Khāzim 1994), 40:18; (Kaʿb b. Mālik al-Anṣārī 1966), 38:8; (Laqīṭ b. Yaʿmur 1971), 3:10; (Ṭufayl al-Ghanawī 1997), 2:20.

24 There are other versions of the poem in which *ka-daʾb* is replaced by words like *ka-umm* ("as the mother of"); see note 4 in ʿAdī b. Zayd al-ʿIbādī ʿAdī b. Zayd 1965, p. 164).

25 The translation is from Suzanne Pinckney Stetkevych (1993, p. 250).

26 In the Qurʾan, it appears only once in the [**Q7**:138]: *ka-mā la-hum ālihatun* (like unto the gods they have). In the poems, there are three appearances: ʿAbīd b. al-Abraṣ (Abīd b. al-Abraṣ and ʿĀmir b. al-Ṭufayl 1913), 41:3 [*illā ka-mā laylata l-ṭalqi*] (the cloudy winds leave the land [in a state] similar to [the weather] on a pleasant night); (Adī b. Zayd 1965), 3:13 [*ka-mā bayna l-liḥāʾi ilā l-ʿasībi*] ([I keep the secret] as if it were between the palm's branch and its bark); (Zuhayr b. Abī Sulmā 2004), 23: 16 [*wa-hya ka-mā hiya*] (it is as it was).

27 Short similes in the Qurʾan: Meccan suras: **Q7**:72; **17**:42, 92; **28**:63; **38**:11; **42**:15. Medinian suras: **Q4**:89, 104. In the poems: (Adī b. Zayd 1965), 3:23; 103:18; 136:1; ʿAmr b. Qamīʾa, *Muntahā l-ṭalab* (Ibn Maymūn al-Baghdādī 1999), 15:23; ʿAmr b. Quʾās al-Murādī, *Muntahā l-ṭalab* (Ibn Maymūn al-Baghdādī 1999), 444:14; (ʿAntara b. Shaddād 1992), 42:2; (Al-Aʿshā Maymūn 1969), 2: 79; 23:23; (Aws b. Ḥajar 1979), 12:4; Muʿāwiya b. Mālik b. Jaʿfar, *Mufaḍḍaliyyāt* (Al-Mufaḍḍal al-Ḍabbī 1918), 105:11; al-Muthaqqib al-ʿAbdī, *Mufaḍḍaliyyāt* (Al-Mufaḍḍal al-Ḍabbī 1918), 28:8; (Al-Nābigha l-Dhubyānī 1996), 1:24; Saʿya b. al-ʿUrayḍ al-Yahūdī, *Aṣmaʿiyyāt* (Al-Aṣmaʾī 1993), 22:1; (Al-Shanfarā 1996), 17:57; (Taʾabbaṭa Sharran 1996), 36:9; (Ṭufayl al-Ghanawī 1997), 1:61; 7:3. There are few instances in the poems in which the vehicle is solely a verb with no pronoun in the verse (Aws b. Ḥajar 1979, 8:2; 48:23; Imruʾ al-Qays 2000, 18:10; 72:8; al-Muthaqqib al-ʿAbdī, *Mufaḍḍaliyyāt* (Al-Muthaqqib al-ʿAbdī 1971), 6:22; Al-Nābigha l-Dhubyānī 1996, 1:35; Rāshid b. Shihāb al-Yashkurī, *Mufaḍḍaliyyāt* (Al-Mufaḍḍal al-Ḍabbī 1918), 86:2). Rarely, the vehicle is a pronoun without a verb (Adī b. Zayd 1965, 135:2; Zuhayr b. Abī Sulmā 2004, 23:16).

28 Meccan suras: **Q6**:94, 110, 133; **7**:27, 29; **11**:112; **15**:90; **17**:24, 42, 92; **18**:48; **21**:5; **34**:54; **42**: 15. Medinian suras: **Q2**: 198, 239; **24**: 59; **58**: 5.

29 Meccan suras: **Q6**:20; **7**:51, 138; **11**:38, 109; **12**:6, 64; **17**:7; **21**:104; **28**: 19, 63, 77; **45**:34; **46**:35; **68**:17; **72**:7; **73**:15. Medinian suras: **Q2**:13, 108, 146, 183, 275, 286; **4**:47, 89, 104, 163; **9**:36, 69; **24**:55; **47**:12; **48**:16; **58**:18; **60**:13.

30 **Q2**: 151; **8**:5. There are two other instances in the poems in which the simile particle opens the verse while the tenor is totally absent: al-Muraqqish al-Aṣghar, *Mufaḍḍaliyyāt* (Al-Mufaḍḍal al-Ḍabbī 1918), 55:18; (Imruʾ al-Qays 2000), 17:20. In a few other

cases in the poems, the simile particle opens the verse with the tenor appearing in previous verses: (Al-Aʿshā Maymūn 1969), 4:3; 5:4; 14:39; ʿAbīd b. al-Abraṣ (Abīd b. al-Abraṣ and ʿĀmir b. al-Ṭufayl 1913), 41:2. (The simile refers to the story of the ancient tribe of ʿĀd, also mentioned in the Qurʾan.). Imruʾ al-Qays b. Jabala, *Muntahā l-ṭalab* (Ibn Maymūn al-Baghdādī 1999), 462:44; (Al-Nābigha l-Dhubyānī 1996), 40:7; Ṣuḥayr b. ʿUmayr, *Aṣmaʿiyyāt* (Al-Aṣmaʿī 1993), 9:22; (Ṭarafa b. al-ʿAbd 2003), 28:14; (Zuhayr b. Abī Sulmā 2004), 9:23; 12:13.

31 See the two translations at https://www.alim.org/quran/compare/surah/2/151/ (accessed on 9 October 2023).

32 See also **Q28**:77 (Meccan); **58**:18 (Medinian).

33 In the Qurʾan, the word *ghuṣṣa* is used in **Q73**:13, not as part of a simile, to describe the conditions in which the unbeliever will live after the Day of the Judgement. It is, however, used literally rather than metaphorically as in ʿAntara's poem, as "choking food". Similar instances in the poems with the same grammatical structure appear in (Antara b. Shaddād 1992), 153:7; (Aws b. Ḥajar 1979), 5:13; al-Ḥārith b. Ẓālim, *Mufaḍḍaliyyāt* (Al-Mufaḍḍal al-Ḍabbī 1918), 88:6; (Imruʾ al-Qays 2000), 1:54; 82:2; (Ṭarafa b. al-ʿAbd 2003), 19:13; (Al-Muthaqqib al-ʿAbdī 1971), 1:28; ʿIlbāʾ b. Arqam, *Aṣmaʿiyyāt* (Al-Aṣmaʿī 1993), 55:15.

34 Other instances are found in (Aws b. Ḥajar 1979), 1:21; (Imruʾ al-Qays 2000), 16:9; Khidāsh b. Zuhayr, *Muntahā l-ṭalab* (Ibn Maymūn al-Baghdādī 1999), 464:35; Muʿāwiya b. Mālik, *Mufaḍḍaliyyāt* (Al-Mufaḍḍal al-Ḍabbī 1918), 105:2, 7.

35 Other examples appear in **Q7**:51 (Meccan) and (Al-Aʿshā Maymūn 1969), 35:19; ʿAwf b. ʿAṭiyya, *Muntahā l-ṭalab* (Ibn Maymūn al-Baghdādī 1999), 49:27; ʿAwf b. ʿAṭiyya, *Mufaḍḍaliyyāt* (Al-Mufaḍḍal al-Ḍabbī 1918), 124:38; (Taʾabbaṭa Sharran 1996), 27:5.

36 In poetry it appears in Aʿshā Bāhila, *Aṣmaʿiyyāt* (Al-Aṣmaʿī 1993), 24:33; (Al-Aʿshā Maymūn 1969), 9:19; 23:21; 33:50, 55; 38:14; (ʿAdī b. Zayd 1965), 25:4; 104:1; al-Akhnas b. Shihāb al-Taghlibī, *Mufaḍḍaliyyāt* (Al-Mufaḍḍal al-Ḍabbī 1918), 41:1, 2; ʿAlqama b. ʿAbada, *Mufaḍḍaliyyāt* (Al-Mufaḍḍal al-Ḍabbī 1918), 119:28; ʿAmr b. Qamīʾa, *Muntahā l-ṭalab* (Ibn Maymūn al-Baghdādī 1999), 11:16; (ʿAntara b. Shaddād 1992), 39: 8; al-Aswad b. Yaʿfur, al-Nahshalī, *Muntahā l-ṭalab* (Ibn Maymūn al-Baghdādī 1999), 52: 3; (Aws b. Ḥajar 1979), 8:8; Bayhas b. ʿAbd al-Ḥārith, *Muntahā l-ṭalab* (Ibn Maymūn al-Baghdādī 1999), 484:34; Bishr b. Abī Khāzim, *Mufaḍḍaliyyāt* (Al-Mufaḍḍal al-Ḍabbī 1918), 96:17; 97:32; Ḍamra b. Ḍamra, l-Nahshalī, *Mufaḍḍaliyyāt* (Al-Mufaḍḍal al-Ḍabbī 1918), 93:9; Abū Duʾād al-Iyādī, *Aṣmaʿiyyāt* (Al-Aṣmaʿī 1993), 65:7; Ḥājib b. Ḥabīb al-Asadī, *Mufaḍḍaliyyāt* (Al-Mufaḍḍal al-Ḍabbī 1918), 111:11; (Al-Ḥārith b. Ḥilliza 1994), 5:7; al-Ḥārith b. Ẓālim, *Mufaḍḍaliyyāt* (Al-Mufaḍḍal al-Ḍabbī 1918), 89:5; (Imruʾ al-Qays 2000), 2:31; 36: 23; Imruʾ al-Qays b. Jabala, *Muntahā l-ṭalab* (Ibn Maymūn al-Baghdādī 1999), 462:38; Muʿaqqir b. Ḥimār al-Bāriqī, *Muntahā l-ṭalab* (Ibn Maymūn al-Baghdādī 1999), 447:4; 448:28; al-Muthaqqib al-ʿAbdī, *Mufaḍḍaliyyāt* (Al-Mufaḍḍal al-Ḍabbī 1918), 28:15; (Al-Nābigha l-Dhubyānī 1996), 1:24; (Abū Qays Ṣayfī b. al-Aslat 1973), 9:6; Salama b. al-Khurshub al-Anmārī, *Muntahā l-ṭalab* (Ibn Maymūn al-Baghdādī 1999), 116:1; (Al-Shanfarā 1996), 17:41; (Taʾabbaṭa Sharran 1996), 23:2; (Ṭarafa b. al-ʿAbd 2003), 1:5; 8:1, 70; ʿUbayd b. ʿAbd al-ʿUzzā, *Muntahā l-ṭalab* (Ibn Maymūn al-Baghdādī 1999), 451:29; (Zuhayr b. Abī Sulmā 2004), 9:5, 33; 12:5; 42:5.

37 (Al-Aʿshā Maymūn 1969), 65:5; (ʿAdī b. Zayd 1965), 161:1, 15; (Bishr b. Abī Khāzim 1994), 11:9; Bishr b. Abī Khāzim, *Mufaḍḍaliyyāt* (Al-Mufaḍḍal al-Ḍabbī 1918), 98:15; (Imruʾ al-Qays 2000), 33:14; al-Mufaḍḍal al-Nukrī, *Aṣmaʿiyyāt* (Al-Aṣmaʿī 1993), 69:5. In the Qurʾan it appears in two Meccan suras: **Q21**:5; **42**:15.

38 Such as *ka-mā jurra l-faṣīlu l-muqarraʿu* ([They drag it] as they drag an ill young camel) (Aws b. Ḥajar 1979, 28: 11). See another example in (Imruʾ al-Qays 2000), 85:4.

39 Such as *ka-mā yurjā l-dunuwwu mina l-biʿād* (as the distant is wished to be close) (ʿAntara b. Shaddād 1992, 42:5). See also al-Aswad b. Yaʿfur al-Nahshalī *Muntahā l-ṭalab* (Ibn Maymūn al-Baghdādī 1999), 52:34; (Aws b. Ḥajar 1979), 48:13; Bishr b. Abī Khāzim, *Muntahā l-ṭalab* (Ibn Maymūn al-Baghdādī 1999), 98:9. In the Qurʾan such as *ka-mā suʾila Mūsā min qablu* (as Moses was questioned before) [**Q2**:108, 183 (Medinian)]. See the other example in **Q58**:5 (Medinian).

40 *Mithl*: Meccan suras: **Q6**:160; **7**:169; **10**:38, 102; **11**:13, 27; **14**:10, 11; **17**:99; **18**:110; **20**:58; **21**:3, 84; **23**: 24, 33; 34, 47; **26**:154, 186; **35**:14; **36**:15, 81; **38**:43; **39**:47; **40**:30, 40; **41**:6, 13; **42**:40; **51**:59; **52**:34; **89**: 8. Although some verses in **Q13** are considered Medinian, most are Meccan—among them Q13:17, 18 in which the *mithl* simile appears. See Theodor Nöldeke (2004, pp. 146-8). Medinian suras: **Q2**:106, 113, 118, 233, 275; **Q3**:140; **4**:11, 140, 176; **5**:31, 36; **8**:31; **14**:10, 11; **65**:12. *Mithl mā*: Meccan suras: **Q6**:93, 124; **23**:81; **28**:48, 79; **51**:23. Medinian suras: **Q3**:73; **5**:95; **11**:89; **60**:11.

41 This verse from the poetry of Al-Aʿshā Maymūn was quoted in different sources to explain another qurʾanic verse [**Q9**:103; Medinian] in which the verb *ṣallā* has the same meaning as in the poetry verse ("to wish"); see Abū ʿUbayda l-Shaybānī (1962, p. 268).

42 **Q5**:36; **10**:102; **11**:27; **14**:11; **16**:126; **17**:99; **18**:110; **20**:58; **22**:60; **23**:24, 47; **26**:154, 186; **36**:15, 47, 81; **39**:47; **40**:31, 40; **41**:6; **42**:40.

43 Meccan suras: **Q6**:93, 124; **7**:169; **10**:27, 38; **11**:13, 89; **13**:17, 18; **14**:10; **21**:3; **28**:48, 79; **42**:11; **46**:10. Medinian suras: **Q2**:113, 118, 194, 228, 233; **3**:13, 73; **4**:11, 140, 176; **5**:195; **8**:31; **60**:11; **65**:12.

44 **Q21**:84; **23**:33, 34, 81; **24**:17; **36**:42; **37**:61; **38**:43; **40**:30; **51** 23, 59; **52**:34; **89**:8.

45 Meccan suras: **Q18**:109; **35**:14; **41**:13. Medinian suras: **Q5**:31.

46 **Q6**:160; **17**:88. All Meccans.

47 ʿĀmir al-Muḥāribī, *Mufaḍḍaliyyāt* (Al-Mufaḍḍal al-Ḍabbī 1918), 91:29; (Al-Aʿshā Maymūn 1969), 3:39; 6:63; 8:4; 10:5; 13:30; 17:18; 18:34; 19:17; 23:17; 25:12; 29:4; 32:35; 34:3, 5; 39:25; 55:41; 56:26; 65:22; 66:2, 8; ʿAbdallāh b. Thawr, *Muntahā l-ṭalab* (Ibn Maymūn al-Baghdādī 1999), 467:17; ʿAbīd b. al-Abraṣ (Abīd b. al-Abraṣ and ʿĀmir b. al-Ṭufayl 1913), 30:35; (ʿAdī b. Zayd 1965), 16:42; al-Akhnas b. Shihāb al-Taghlibī, *Mufaḍḍaliyyāt* (Al-Mufaḍḍal al-Ḍabbī 1918), 41:20; ʿAlqama b. ʿAbada, *Mufaḍḍaliyyāt* (Al-Mufaḍḍal

al-Ḍabbī 1918), 120:16; ʿĀmir b. al-Ṭufayl (Abīd b. al-Abraṣ and ʿĀmir b. al-Ṭufayl 1913), 2:5; 10:3, 4; 11:11, 12; 37:3; 50:1; (Antara b. Shaddād 1992), 10:4; 11:11; 26:23; 46:8, 10; 48: 5; 69:9; 81:12; 82:4; 94:7; 98:3; 109:13; 110:11; 121:3; 138:2; 141:10, 13; 145:9; al-Aswad b. Yaʿfur al-Nahshalī, *Muntahā l-ṭalab* (Ibn Maymūn al-Baghdādī 1999), 56:12; (Aws b. Ḥajar 1979), 49:1; Dawsar b. Dhuhayl al-Qurayʿī, *Aṣmaʿiyyāt* (Al-Aṣmaʿī 1993), 50:6, 10; al-Find al-Zimmānī, *Muntahā l-ṭalab* (Ibn Maymūn al-Baghdādī 1999), 477:65; 479:97, 20; (Al-Ḥārith b. Ḥilliza 1994), 69:68, 81; al-Ḥārith b. Waʿla l-Jarmī, *Mufaḍḍaliyyāt* (Al-Mufaḍḍal al-Ḍabbī 1918), 32:2; Ḥājiz b. ʿAwf, *Muntahā l-ṭalab* (Ibn Maymūn al-Baghdādī 1999), 453:3; al-Ḥusayn b. al-Ḥumām al-Murrī, *Mufaḍḍaliyyāt* (Al-Mufaḍḍal al-Ḍabbī 1918), 12:17; ʿIlbāʾ b. Arqam, *Aṣmaʿiyyāt* (Al-Aṣmaʿī 1993), 56:4; (Imruʾ al-Qays 2000), 1:16, 39; 2:14; 30:5; 53:22; 76:10; (Kaʿb b. Mālik al-Anṣārī 1966), 6:8; 13:6; 32:11; 66:16; Kaʿb b. Saʿd al-Ghanawī, *Muntahā l-ṭalab* (Ibn Maymūn al-Baghdādī 1999), 350:26; Khidāsh b. Zuhayr, *Muntahā l-ṭalab* (Ibn Maymūn al-Baghdādī 1999), 464:39; Maqqās al-ʿĀidhī, *Mufaḍḍaliyyāt* (Al-Mufaḍḍal al-Ḍabbī 1918), 84:4; Muʿādh b. Muʿāwiya b. Jaʿfar, *Mufaḍḍaliyyāt* (Al-Mufaḍḍal al-Ḍabbī 1918), 105:15; Muʿaqqir b. Ḥimār, *Muntahā l-ṭalab* (Ibn Maymūn al-Baghdādī 1999), 448:2; al-Muraqqish al-Aṣghar, *Mufaḍḍaliyyāt* (Al-Mufaḍḍal al-Ḍabbī 1918), 55:14; (Al-Mutalammis al-Ḍubaʿī 1970), 5:10; (Al-Nābigha l-Dhubyānī 1996), 3:12; Qays b. al-Khaṭīm, *Muntahā l-ṭalab* (Ibn Maymūn al-Baghdādī 1999), 342 2; 343:11; 344:5; al-Samawʾal b. ʿĀdiyāʾ, *Muntahā l-ṭalab* (Ibn Maymūn al-Baghdādī 1999), 433:4; Suʿdā bint al-Shamardal al-Juhaniyya, *Aṣmaʿiyyāt* (Al-Aṣmaʿī 1993), 27:2; (Taʾabbaṭa Sharran 1996), 12:4; 15:9; 56:3; 61:3; (Ṭarafa b. al-ʿAbd 2003), 8:40; 17:37; (Ṭufayl al-Ghanawī 1997), 1:62, 65; 5:21; ʿUrwa b. al-Ward, *Muntahā l-ṭalab* (Ibn Maymūn al-Baghdādī 1999), 145:6; (Zuhayr b. Abī Sulmā 2004), 2:41; 14:14; 15:13; 48:3.

48    Meccan suras: **Q6**:160 (twice); **7**:169; **10**:27; **10**:38; **11**:27; **13**:17, 18; **14**:10, 11; **17**:88, 99; **18**:110; **20**:58; **21**:3, 84; **23**:24, 33, 34, 47; **26**:154, 186; **36**:15, 81; **38**:43; **40**:40; **41**:6; **42**:40; **46**:10; **52**:34; **89**:8. Medinian suras: **Q2**:106; **3**:13, 140, 165; **4**:140; **5**:36; **24**:17; **65**:12.

49    Meccan suras: **Q6**:93, 124; **11**:89; **16**:126; **22**:60; **23**:81; **28**:48, 79. Medinian suras: **Q2**:137, 194; **3**:73; **5**:95; **60**:11.

50    (Al-Aʿshā Maymūn 1969), 36:15; 36:49; 41:5; 65:5; 78:10, 21; ʿAbīd b. al-Abraṣ (Abīd b. al-Abraṣ and ʿĀmir b. al-Ṭufayl 1913), 19:24; 23:23; 29:12; (ʿAdī b. Zayd 1965), 35:1; 72:7; 103:5; (ʿAntara b. Shaddād 1992), 11:22; 38:10; 49:10; 59:5; (Aws b. Ḥajar 1979), 14:11; (Bishr b. Abī Khāzim 1994), 38:33; al-Find al-Zimmānī, *Muntahā l-ṭalab* (Ibn Maymūn al-Baghdādī 1999), 477:56, 64; Ḥājiz b. ʿAwf al-Asadī, *Muntahā l-ṭalab* (Ibn Maymūn al-Baghdādī 1999), 453:21; (Zuhayr b. Abī Sulmā 2004), 242.

51    The translation is from Kirill Dmitriev (2009, p. 358).

52    See other examples in: (Al-Aʿshā Maymūn 1969), 30:27; 52:36; ʿAbīd b. ʿAbd al-ʿUzzā, *Muntahā l-ṭalab* (Ibn Maymūn al-Baghdādī 1999), 452:7; ʿAbīd b. al-Abraṣ (Abīd b. al-Abraṣ and ʿĀmir b. al-Ṭufayl 1913), 11:26; (ʿAntara b. Shaddād 1992), 57:3; 131:15; Ḥājib b. Ḥabīb al-Asadī, *Mufaḍḍaliyyāt* (Al-Mufaḍḍal al-Ḍabbī 1918), 111: 3; al-Jumayḥ al-Asadī, *Aṣmaʿiyyāt* (Al-Aṣmaʿī 1993), 80: 13; (Ṭufayl al-Ghanawī 1997), 2:20; (Zuhayr b. Abī Sulmā 2004), 17:6; Zuhayr b. Masʿūd, *Muntahā l-ṭalab* (Ibn Maymūn al-Baghdādī 1999), 474:15.

53    Meccan suras: **Q18**:109, 110; **21**:84; **23**:24, 33; **38**:43; **39**:47; **41**:6; **42**:11. Medinian suras: **Q3**:13; **5**:36, 95; **13**:18.

54    (Al-Aʿshā Maymūn 1969), 29:21; ʿAbīd b. al-Abraṣ (Abīd b. al-Abraṣ and ʿĀmir b. al-Ṭufayl 1913), 26:19; (ʿAntara b. Shaddād 1992), 45:6; ʿAlqama b. ʿAbada, *Mufaḍḍaliyyāt* (Al-Mufaḍḍal al-Ḍabbī 1918), 119:37; (Imruʾ al-Qays 2000), 72:22; (Kaʿb b. Mālik al-Anṣārī 1966), 26:10; 31: 1; (Zuhayr b. Abī Sulmā 2004), 45:2.

55    (Al-Aʿshā Maymūn 1969), 1:59; 3:46; 5:26; 12:17; 13:45; 23:12; 29:36; 33:11; 34:36; 38:24; 46:3; 65: 9; 71:3; 77:13; 78:27; ʿAbīd b. al-Abraṣ (Abīd b. al-Abraṣ and ʿĀmir b. al-Ṭufayl 1913), 24:11; 25:3; (ʿAdī b. Zayd 1965), 4:22; 49:1; ʿĀmir b. al-Ṭufayl (Abīd b. al-Abraṣ and ʿĀmir b. al-Ṭufayl 1913), 40:5; (ʿAmr b. Qamīʾa 1919), 5:11; (ʿAntara b. Shaddād 1992), 1:2; 9:16; 29:1; 39:20; 64:4; 81:1; 107:18; 111:8; 112:5; 146:5; 147:5; 151:1; 154:17; al-Aswad b. Yaʿfur al-Nahshalī, *Muntahā l-ṭalab* (Ibn Maymūn al-Baghdādī 1999), 52:19; (Aws b. Ḥajar 1979), 41:3; 48:17; (Bishr b. Abī Khāzim 1994), 40:6; Bishr b. Abī Khāzim, *Muntahā l-ṭalab* (Ibn Maymūn al-Baghdādī 1999), 98: 3; Dhū l-Iṣbaʿ al-ʿAdwānī, *Muntahā l-ṭalab* (Ibn Maymūn al-Baghdādī 1999), 122: 14; Ḥājib b. Ḥabīb, *Aṣmaʿiyyāt* (Al-Aṣmaʿī 1993), 82:3; (Imruʾ al-Qays 2000), 3:14; 16:7; 35:5; 36:30; 71:19; 74:11; 79:2; (Kaʿb b. Mālik al-Anṣārī 1966), 7:4; 26:10; Khidāsh b. Zuhayr, *Muntahā l-ṭalab* (Ibn Maymūn al-Baghdādī 1999), 464:42; al-Munakhkhil al-Yashkurī, *Aṣmaʿiyyāt* (Al-Aṣmaʿī 1993), 14:8; al-Muraqqish al-Akbar (Al-Muraqqishān 1998), 14:22; Abū Qays Ṣayfī b. al-Aslat, *Muntahā l-ṭalab* (Ibn Maymūn al-Baghdādī 1999), 445: 11; (Ṭarafa b. al-ʿAbd 2003), 8:30; Thaʿlaba b. Ṣuʿayr, *Mufaḍḍaliyyāt* (Al-Mufaḍḍal al-Ḍabbī 1918), 24:22; (Zuhayr b. Abī Sulmā 2004), 18:3; 32:12; 53:26.

56    (Al-Aʿshā Maymūn 1969), 2:42; 4:62; 6:31; 11:29; 39:40; 40:18; 65:7; 79a:6, 8; ʿAbīd b. al-Abraṣ (Abīd b. al-Abraṣ and ʿĀmir b. al-Ṭufayl 1913), 20:2; 24:19; (ʿAdī b. Zayd 1965), 16:8; 118:3; 138:8; al-Akhnās b. Shihāb, *Muntahā l-ṭalab* (Ibn Maymūn al-Baghdādī 1999), 180:25; (Alqama b. ʿAbada 1993), 3: 25; ʿĀmir b. al-Ṭufayl (Abīd b. al-Abraṣ and ʿĀmir b. al-Ṭufayl 1913), 8: 2; 11: 6; (ʿAmr b. Qamīʾa 1919), 11: 16; (ʿAntara b. Shaddād 1992), 13:1; 30:6; 60:4; 93:16; 107:25; 116:2; 118:6; 130:36; 153:4; 164:5; al-Asʿar al-Juʿfī, *Aṣmaʿiyyāt* (Al-Aṣmaʿī 1993), 44:11; ʿAwf b. ʿAṭiyya, *Mufaḍḍaliyyāt* (Al-Mufaḍḍal al-Ḍabbī 1918), 124:14; Aws b. Ḥajr, 2:9; 5:3; 14:8, 10; 35:14; (Bishr b. Abī Khāzim 1994), 29:17; 40:33; 41:5; 46:14; Bishr b. Abī Khāzim, *Muntahā l-ṭalab* (Ibn Maymūn al-Baghdādī 1999), 98:18; 102:14; Ḥājiz b. ʿAwf, *Muntahā l-ṭalab* (Ibn Maymūn al-Baghdādī 1999), 453:9; 454:31; (Imruʾ al-Qays 2000), 23:1; 72:12; Khidāsh b. Zuhayr, *Muntahā l-ṭalab* (Ibn Maymūn al-Baghdādī 1999), 463:4; (Al-Shanfarā 1996), 17:51; (Taʾabbaṭa Sharran 1996), 13:2; (Ṭarafa b. al-ʿAbd 2003), 26:13; (Ṭufayl al-Ghanawī 1997), 2:29, 37; (Zuhayr b. Abī Sulmā 2004), 23:19. There are other instances in which the double annexation is followed by a word which is part of the vehicle.

57    The single-noun vehicle appears in two Meccan suras **Q17**:88; **35**:14. The annexation appears in Meccan suras: **Q40**:30; **41**:13; **51**:59, and Medinian suras **Q2**:113; **4**:11, 176.

58　　Meccan: **Q18**:18; **27**:44; **76**:19. Medinian: **Q24**: 39. In poetry: (Al-Aʿshā Maymūn 1969), 27:6; 34:17; (ʿAntara b. Shaddād 1992), 93:7; Al-Ḥārith b. Ḥilliza, *Mufaḍḍaliiyāt* (Al-Mufaḍḍal al-Ḍabbī 1918), 62:8; al-Muraqqish al-Aṣghar (Al-Muraqqishān 1998), 4:5; (Ṭarafa b. al-ʿAbd 2003), 21:9; (Al-Nābigha l-Dhubyānī 1996), 4:28; (Zuhayr b. Abī Sulmā 2004), 11:2; 21:6.

59　　This appears only once in the Qurʾan in **Q27**:44. In poetry it is seen in (Al-Aʿshā Maymūn 1969), 16:32; 19:2; 23:4; (ʿAntara b. Shaddād 1992), 30:11; 98:6; 135:10; (Aws b. Ḥajar 1979), 32:9; (Ṭufayl al-Ghanawī 1997), 8:18; (Zuhayr b. Abī Sulmā 2004), 1:58.

60　　(ʿAdī b. Zayd 1965), 153:1; ʿĀmir b. Juwayn, *Muntahā l-ṭalab* (Ibn Maymūn al-Baghdādī 1999), 485:20; (Imruʾ al-Qays 2000), 72:1; Yazīd b. Khaddhāq al-Shannī, *Mufaḍḍaliyyāt* (Al-Mufaḍḍal al-Ḍabbī 1918), 78:7; 79:10.

61　　In Arabic, *lammā* (when) is considered a conditional particle.

62　　Short similes in the Qurʾan appear in **Q2**:273 [Medinian]; **18**:18 [Meccan]; **27**:44, 88 [Meccan]. In poetry, they are seen in ʿĀmir b. Juwayn, *Muntahā l-ṭalab* (Ibn Maymūn al-Baghdādī 1999), 485:20; (Al-Aʿshā Maymūn 1969), 16:32; 27:6; 34:17; 36:33; 55:2; (ʿAdī b. Zayd 1965), 9:8; 152:2; 153:1; (ʿAntara b. Shaddād 1992), 114:26; (Bishr b. Abī Khāzim 1994), 46:24; (Imruʾ al-Qays 2000), 72:1; 77:27; (Kaʿb b. Mālik al-Anṣārī 1966), 61:7; Rāshid b. Shihāb al-Yashkurī, *Mufaḍḍaliyyāt* (Al-Mufaḍḍal al-Ḍabbī 1918), 87:7; (Ṭarafa b. al-ʿAbd 2003), 17:12, 76; 21:9; (Zuhayr b. Abī Sulmā 2004), 1:58; 7:32.

63　　Other prolonged similes in poetry are in: (Al-Aʿshā Maymūn 1969), 23:4; 25:8; 36:42; (ʿAntara b. Shaddād 1992), 30:11; 93 7; 98:6; (Aws b. Ḥajar 1979), 32:9; al-Muraqqish al-Aṣghar (Al-Muraqqishān 1998), 4:5; (Al-Nābigha l-Dhubyānī 1996), 4:28; (Ṭufayl al-Ghanawī 1997), 8:17, 18; (Zuhayr b. Abī Sulmā 2004), 21:6.

64　　**Q24**:39. In poetry: Yazīd b. Khaddhāq al-Shannī, *Mufaḍḍaliyyāt* (Al-Mufaḍḍal al-Ḍabbī 1918), 78:7; 79:10.

65　　(Al-Aʿshā Maymūn 1969), 19:2; (ʿAntara b. Shaddād 1992), 135:10; Al-Ḥārith b. Ḥilliza, *Mufaḍḍaliiyāt* (Al-Mufaḍḍal al-Ḍabbī 1918), 62: 8.

66　　*Taḥsibu/taḥsabu* is found in **Q18**:18; **27**:88; and (Al-Aʿshā Maymūn 1969), 19:2; 25:8; 27:6; 34:17; 55: 2; (ʿAdī b. Zayd 1965), 9:8; 152:2; (ʿAntara b. Shaddād 1992), 114:26; 135:10; Al-Ḥārith b. Ḥilliza, *Mufaḍḍaliiyāt* (Al-Mufaḍḍal al-Ḍabbī 1918), 62: 8; (Imruʾ al-Qays 2000), 77: 27; (Kaʿb b. Mālik al-Anṣārī 1966), 61: 7; (Ṭarafa b. al-ʿAbd 2003), 17:12, 76; 21:9; (Zuhayr b. Abī Sulmā 2004), 1:58. *yaḥsibu/yaḥsabu* appears in **Q2**:273; **24**:39; and in (Al-Aʿshā Maymūn 1969), 23:4; (ʿAntara b. Shaddād 1992), 93:7; al-Muraqqish al-Aṣghar (Al-Muraqqishān 1998), 4:5; (Zuhayr b. Abī Sulmā 2004), 11:2; 21:6.

67　　(Al-Aʿshā Maymūn 1969), 32:44; 55:37; (Aws b. Ḥajar 1979), 26: 8; (Bishr b. Abī Khāzim 1994), 23: 3; Bishr b. Abī Khāzim, *Munhā l-ṭalab* (Ibn Maymūn al-Baghdādī 1999), 101:10.

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
