# Peer review of "Poetry and the Qurʾan: The Use of tashbīh Particles in Classical Arabic Texts"

_religions, doi:10.3390/rel14101326_

Round 1

Reviewer 1 Report

This paper is an important contribution to our understanding of the Qur'an's relationship to early Arabic poetry; a subject all-too-often overlooked in modern scholarship. The author presents a very detailed exploration of the use of simile in both corpora. In general, I agree with the author's conclusions. I have a few comments, which I hope the author may find useful:

1. I am not altogether convinced of the significance of some of the structural similarities and locations of similes in verses. First, there seems in general to be quite a wide range of structural possibilities for most of the similes and while there are some shared characteristics, I am not sure if pointing out statistically quite insignificant cases (e.g., one or two per corpora) is useful to the overall argument. It does make the article seem very comprehensive, but I wonder if some of these might be best left to footnotes. Do these really tell us anything about the relationship of the Qur'an to poetry? Common examples = yes, minor examples = unlikely. Perhaps such similarities are merely a fact of Arabic syntax?

Regarding location of similes in verses, again some of these seems to be statistically insignificant. While poetic verses are constrained by rhyme and form, Qur'anic verses are not and vary considerably in length, so I am not sure how useful this comparison is. Again, worth pointing out some common occurrences (which are interesting), but perhaps not every overlap.

2. I found myself wondering at the beginning of the article how the author viewed the relationship of the Qur'an to pre-Islamic poetry in terms of reliance on/influence of, etc. This is glossed over on pg. 6 (e.g., line 151) by use of the word 'reflects'. Also, the temporal relationship seems unclear when the author states, on the same page, that a poetic line 'echo[s] qur’anic verses' (line 148). The issue is, however, very succinctly cleared up on page 10 (line 295>). As such, it might be useful to detail some of the points in this important paragraph a little earlier in the article.

3. Page 9, first paragraph, includes some quite general conclusions. This seems a little odd given that it is only halfway through the article. This might be because the ka- particle accounts for most of the shared content between the two corpora, but it reads a little premature.

4. On the above point - it would be very useful to have a graph, as those in Figures 1-3, that clearly shows the most common shared simile types between the two corpora. This can be gleaned from Figure 3, but a separate graph would be helpful for the reader.

5. In the references to previous studies comparing the Qur'an and poetry (pg. 2), Sinai's Rain-Giver, Bone-Breaker, Score-Settler (American Oriental Series, 2019) is missing, which is quite a significant recent contribution.

6. Line 97 has an incorrect Qur'anic reference: 2:74 should be 3:49.

7. The first paragraph describing the statistical distribution of the ka- simile is a little hard to follow, especially the second sentence. Consider revising.

9. Percentages are generally used throughout, but whole numbers might illustrate the relative occurrences of each term clearer when comparing the corpora in some sections.

10. I would precede the first footnote with an explanation that the Digital Humanities Quarterly article clearly outlines how the similes were located in the corpus of texts, as well as describing the generation of the corpus itself. As this isn't detailed in the article itself and it is an obvious question for the reader.

11. Following Bauer's so-called 'negative intertextuality' line of thought, is there anything to say on the types of simile that don't occur in the two corpora?!

Once again, I found this an excellent and much needed article, which clearly demonstrates the utility of comparing the two corpora.

Overall the English is excellent. A couple of minor points:

Line 218: 'deserving of greater light shed' (incorrect syntax) - perhaps 'a unique contruction, deserving of further illumination'.

Line 262-3. The two sentences, beginning with 'The vehicle is rarely...', should be either linked with a semi colon or a compound sentence. Otherwise the close relationship of the second sentence to the first is lost and throws the reader.

Reviewer 2 Report

Hats off to the author(s) for synthesizing such a large amount of difficult material. It is perhaps not surprising that pre-Islamic poetry (PIP) and the Quran (Q) should use many of the same devices. But the change across the Meccan and Medinan surahs appears significant. To better bring out that significance, it may be useful to clear up some points of method.

Certain assumptions should be spelled out even if they are widely shared––esp. because the findings may be relevant in confirming those assumptions (see below). The first is that PIP is in fact pre-Islamic. The opposite view, that all of it is a later fabrication, seems unlikely; but it is indisputable that the versions available to us today are the result of centuries of sifting, tweaking, suppression, and substitution. Since the original poems (if such existed) are now beyond retrieval, we have no choice but to work with what we have. Therefore, no single data point should be given much weight. Very frequent correlations, however, are probably significant.

Similarly, the authors accept the traditional dating of the Q (which has been independently confirmed in Sadeghi and Bergmann, "Codex").

Finally, they accept the traditional division into Meccan and Medinan surahs, on which see the study by Sadeghi, which summarizes the state of the field with respect to the ordering of the verses).

I have no objection to any of these assumptions, but they should be stated at the outset.

(Off topic but important: you might point out that the REI is not a publicly accessible database. At least, I couldn't find it. Normally, one cites poems from their diwans, not from a database that is limited-access. I realize that wouldn't be practical here, but is there some way readers might be able to use the REI?

And, since we're talking about the database: Are there no correlations with poetry by women poets such as al-Khansa and Layla al-Akhyaliyyah? As I recall, al-Khansa at least does use tashbih mursal.)

With respect to similes, the study limits itself to the tashbih mursal, which is perfectly understandable for practical reasons. But without knowing how many other kinds of similes there are, and how frequent they are with respect to the mursal, it is difficult to assess how representative the findings are. It would probably suffice to say that the mursal is very frequent, and so its behavior is significant, while acknowledging that there are many other kinds of tashbih that remain to be investigated.

Tangential point of method: Line 297 misstates what is meant by a formula. Basically, it's a template that allows a poet to retain the content of a verse while varying its meter and rhyme. The standard exposition is Lord and Parry, Singer of Tales, and there are several studies--notably Zwettler's––that explore how well that paradigm fits PIP. Spoiler alert: not very well.

In the discussion, the essay vacillates between describing what texts do and assuming a kind of historical progression. This wanting-it-both-ways comes through in the verb tenses. In 527-28, "the Q shares," present tense. But in 530-31, "these features faded away," past tense. In my view, the present tense is always the right one for talking about what texts do. Once you've established that, you can write a separate sentence or paragraph saying: "Given these correlations, we can postulate the following historical progression. First A happened, then B happened," etc., all past tense.

BOTTOM LINE: If you do it this way, you can show how your data actually helps *confirm* certain disputed points like the ancientness of PIP and the ordering of the verses of the Q. This is significant! After that you can go on and sketch out the possible historical development of the Q away from PIP. This is new, and it's where the essay should lead; it shouldn't be mentioned offhand here and there.

The English is perfectly clear; the following seem to be oversights, or editorial choices that might be re-thought.

Using the same notation for words like ka- and for roots like ḥsb is confusing. Ka- is a word but ḥsb is not. On first use, try "derivatives of the root ḤSB" (in small caps?) and "ḤSB-derivatives" thereafter. Same with SHBH.

"Thought though," whch appears several times (incl. 504, makes no sense in English. A one-word equivalent for ḤSB might be "deem."

Line 241: "Umm al-Rabāb" is missing from the translation.

Line 263-64: "No observed similarities" is confusing since the text goes on to mention what appears to be a similarity.

Line 279: "even if it was just as you said" would be a better translation.

404: al-A`shā: no italics

406: Al-A`sha: first A should be lower case.

415: times occasions

Reviewer 3 Report

I read the paper with significant interest, and I find the innovative use of the comparative stylistic methodology commendable.  However, I must point out that the conclusion is somewhat misleading, particularly in reference to the sections found on pages 18-19. " Investigating the sources of these “Medinian” features is beyond the scope of this article, but a comparative study between them and those in other texts known during the Medinian period (such as the Bible) may be useful. Should such studies prove that Medinian similes are closer in structure and possibly in lexica to biblical texts, this would demonstrate that text followed two proto-compositions: one purely Arabic (Arabic poetry, its beginnings) and one non-Arabic (mainly biblical, possibly translated into Arabic in its later Revelations). These two proto-compositions are thus important for new understanding of and new insights into the Arabic scripture." The article does not delve into any stylistic comparisons with the Bible in order to assert the presence of a stylistic influence that their methodology has newly uncovered. Moreover, the author should be aware that their study may be perceived as Eurocentric, primarily due to the methodology they have employed to analyze specific stylistic elements within the texts. Therefore, the author is advised to import some definitions of tashbih from Arabic balaghah sources.  Tashbih is not a modern Euro-American simlie?  Numerous studies exist that deconstruct the linkages between ancient Greek, Latin, and modern European languages concerning philosophical and literary terms. The author should exercise caution when automatically adopting Euro-American terms without elucidating their distinct conceptual differences within Arabic balaghah sources.

The author is strongly encouraged to furnish a comparative example from both the Quran and pre-Islamic poetry, aimed at elucidating their argument concerning influence. Including an example for each type of tashbih would significantly enhance the educational merit of the paper. All the examples should be presented in Arabic script, followed by their English translation. 

The article shouldn't neglect to engage with premodern Arabic balaghah (roughly translated as rhetoric) sources regarding tashbih. In these Arabic sources, tashbih is classified into over 70 distinct types. It would be prudent for the author to ascertain the presence of these 70 types within the texts under scrutiny before applying their digital methodology. By the way, can this digital methodology rediscover instances of tashbih that do not involve using particles? Can the digital methodology rediscovers the 70 types of tashbih in both texts and give us a map of how each text/author employed these types similarly and differently?  Furthermore, the author is encouraged to address the significance of tashbih within Arabo-Islamic culture through a comprehensive literature review at the beginning of the article. This could encompass explorations of how tashbih has engendered contentious debates among scholars of the Quran and hadith, both in modern and premodern eras. In this regard, I recommend the inclusion of two pertinent sources for engagement:

Salama, Mohammad. (2018), /The Qur’an and Modern Arabic Literary Criticism: From Ṭāhā to Naṣr/. (London, Bloomsbury).

Holtzman, L. (2018), /Anthropomorphism in Islam: The Challenge of Traditionalism (700-1350)/(Edinburgh, Edinburgh University Press).

Hence, substantial revisions are necessary for the article to address these queries that inadvertently neglect the contextual relevance of tashbih within premodern Arabic literary criticism.

NA

Round 2

Reviewer 3 Report

The author did not actually engage with any of my questions. Every point I raised, he/she said, will be done in the future. The article, for me, still looks Eurocentric. The author needs to engage with Arabic balaghah sources about tashbih/simlie in the introduction. He just noted one reference without extracting any definition of tasbih mursal.   

"See, for example, the informative details and different sub-types of simile found in Yūsuf b. Abī Bakr al-Sakkākī, Miftāḥ al-ʿulūm, ed. Naʿīm Zarzūr (Beirut: Dār al-Kutub al-ʿIlmiyya, 1987), 3: 332-355" 

In his response, he says "The main type of simile that I refer to (called the tashbīh mursal) is not a Euor-American simile, rather it is taken from the pre-modern Arabic books on rhetoric" and in another place says "I added one of these theoretical pre-modern Arabic sources in footnote 3. However, since the article is about the practical use of one type of simile in the two corpora, it does not deal with the theory of balagha/rhetoric. 

The article requires the inclusion of examples and clear definitions of the specific type of simile being used, aiming to introduce it thoughtfully to English readers. It appears to follow a Eurocentric tradition of scholarship. The author noted the scarcity of articles addressing Arabic literary devices, which is indeed true. I also considered sharing my article and an upcoming book on Arabic "jinas" (wordcraft), but hesitated due to feeling self-conscious about it. If you wish to propose these works to the author, it might prompt a reconsideration of their approach to non-European literary poetics, potentially moving away from Eurocentrism.

Rashwan, Hany. “Arabic jinās is not pun, wortspiel, calembour or paronomasia: A post-Eurocentric comparative approach to the conceptual untranslatability of literary terms in Arabic and ancient Egyptian cultures,” in Rhetorica: A Journal of the History of Rhetoric, 2020, vol. 38 (4): 335–370.
https://doi.org/10.1525/rh.2020.38.4.335

Author Response

Dear Peer Reviewer,

Please accept my heartfelt gratitude for your invaluable feedback. I have conscientiously considered each of your suggestions and incorporated the necessary revisions into this resubmitted version of the article, which are highlighted in turquoise for your convenience. Below, I outline the changes I've made in response to your comments, denoted in red text.

The author did not actually engage with any of my questions. Every point I raised, he/she said, will be done in the future. The article, for me, still looks Eurocentric. The author needs to engage with Arabic balaghah sources about tashbih/simlie in the introduction. He just noted one reference without extracting any definition of tasbih mursal.

The article requires the inclusion of examples and clear definitions of the specific type of simile being used, aiming to introduce it thoughtfully to English readers. It

The article now includes explicit definitions and exemplifications of the specific type of simile under examination, with a thoughtful aim to introduce these concepts to English-speaking readers. I have meticulously defined the two principal forms of "tashbih" featured in my article, particularly emphasizing "tashbih mursal," drawing upon definitions found in Arabic balagha sources. Furthermore, I have traced the evolution of these definitions within medieval Arabic works on rhetoric, conducting an in-depth analysis of the rhetoricians' definitions and the accompanying illustrative examples they provide (see lines 44-140).

Additionally, I have elucidated the meanings of the two terms— "tamthil" and "compound simile"—that appear on pages... These definitions are grounded in medieval Arabic rhetoric texts (see lines 240-261, 276-317).

.... also considered sharing my article and an upcoming book on Arabic "jinas" (wordcraft), but hesitated due to feeling self-conscious about it. If you wish to propose these works to the author, it might prompt a reconsideration of their approach to non-European literary poetics, potentially moving away from Eurocentrism.

The article begins by addressing the challenge of using English terminology to explain Arabic rhetorical devices. I have also incorporated the theory presented in the article, which underscores these concerns (see lines 32-43).

I sincerely apologize for not implementing these changes in my previous revision of the article. I am confident that the newly integrated material has significantly enriched the article and its overall quality.

With warm regards

Round 3

Reviewer 3 Report

The article appears to be in improved condition following the inclusion of definitions and examples. Consequently, the author may consider updating the article's title to incorporate the Arabic term "tashbih" instead of "simile" in order to acknowledge the distinct nature of each literary device and avoid automatic misconceptions.

This is also a good reference you may like to include

Smyth, William. “Some Quick Rules Ut Pictura Poesis: The Rules for Simile in ‘Miftāḥ Al-ʿUlūm.’” Oriens 33 (1992): 215–29. https://doi.org/10.2307/1580605.

Author Response

Dear peer reviewer,

Thank you for your comments.

The new amendments, which are now highlighted in green, include changes to the title and a reference to Smyth's article in footnote 3, alongside my previous reference to al-Sakkaki's Miftāḥ al-ʿulūm.

Best regards.